# Adversarial Flow Models

**Shanchuan Lin**[1]   **Ceyuan Yang**[1]   **Zhijie Lin**[1]   **Hao Chen**[1]   **Haoqi Fan**[1]

## Abstract

We present adversarial flow models, a class of generative models that belongs to both the adversarial and flow families. Our method supports native one-step and multi-step generation and is trained with an adversarial objective. Unlike traditional GANs, in which the generator learns an arbitrary transport map between the noise and data distributions, our generator is encouraged to learn a deterministic noise-to-data mapping. This significantly stabilizes adversarial training. Unlike consistency-based methods, our model directly learns one-step or few-step generation without having to learn the intermediate timesteps of the probability flow for propagation. This preserves model capacity and avoids error accumulation. Under the same 1NFE setting on ImageNet-256px, our B/2 model approaches the performance of consistency-based XL/2 models, while our XL/2 model achieves a new best FID of 2.38. We additionally demonstrate end-to-end training of 56-layer and 112-layer models without any intermediate supervision, achieving FIDs of 2.08 and 1.94 with a single forward pass and surpassing the corresponding 28-layer 2NFE and 4NFE counterparts with equal compute and parameters. The code is available at this repository.

## 1. Introduction

Flow matching (Lipman et al., 2023) is a generative method that has achieved state-of-the-art performance across multiple domains. It frames generation as transporting samples from a prior distribution to the data distribution. A probability flow is established by interpolating between data samples and prior samples, and a neural network learns the gradient field of this flow. At inference time, each sample is transported iteratively by querying the network for gradients, incurring a high computational cost.

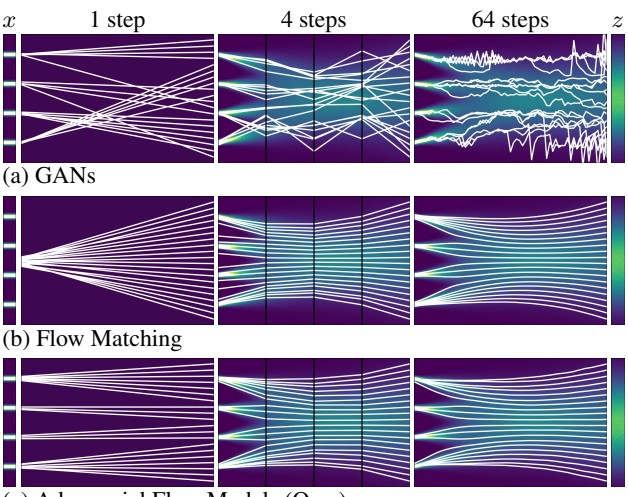

*Figure 1.* Models trained on a 1D Gaussian mixture. (a) GANs learn arbitrary transport maps. (b) Flow matching learns a deterministic transport map but have large discretization errors on low sampling steps. (c) Adversarial flow models support any-step training and generation with a deterministic optimal transport map.

Recent methods accelerate generation by training networks to predict distant positions along the flow rather than instantaneous gradients. This can be achieved either by distilling from a pre-trained flow-matching model (Salimans & Ho, 2022; Liu et al., 2023) or by training from scratch with consistency objectives, forming a new class of generative models (Song et al., 2023). However, even when targeting single-step or few-step generation, consistency-based models must still be trained across all timesteps to propagate consistency. This consumes model capacity and introduces error accumulation. Furthermore, models operating in fewer steps have less capacity to predict the exact transformations of targets produced with more steps, so pointwise matching or even moment-matching losses can lead to some degree of blurriness. For these reasons, many state-of-the-art few-step generation models still rely on distributional matching methods, especially adversarial training, for final refinement (Lin et al., 2025; Chen et al., 2025).

Adversarial training originates from generative adversarial networks (GANs) (Goodfellow et al., 2014). It is itself a standalone class of generative models that supports single-step generation. However, adversarial training from scratch often suffers from stability issues. Recent works explor-

---

[1]ByteDance Seed. Correspondence to: Shanchuan Lin <peterlin@bytedance.com>.

*Proceedings of the 43rd International Conference on Machine Learning*, Seoul, South Korea. PMLR 306, 2026. Copyright 2026 by the author(s).

ing adversarial training from scratch adopt non-standard architectures (Huang et al., 2024; Zhu et al., 2025). Some further rely on additional frozen feature networks (Kang et al., 2023; Hyun et al., 2025). When we switch to a standard transformer architecture (Vaswani et al., 2017), training simply diverges.

As shown in Figure 1, we find that one of the key reasons GANs are difficult to train is that the adversarial objective alone does not define a single optimization target. This differs markedly from other established objectives, such as flow matching, which has a unique ground-truth probability flow determined by the interpolation function, and autoregressive modeling, which has ground-truth token probabilities determined by the training corpus. In GANs, the generator is tasked with transporting samples from the prior to the data distribution, but the adversarial objective only enforces matching between the data distributions without constraining the transport map. Therefore, there are infinitely many valid transport maps that the generator may choose, depending on the weight initialization and the stochastic training process. This creates optimization difficulties because the generator keeps drifting during training.

In this paper, we propose adversarial flow models, a class of generative models that belongs to both the adversarial and flow families. Our models are trained with an adversarial objective, which naturally supports single-step training and generation without consuming model capacity to learn the intermediate timesteps required by consistency methods. At the same time, they belong to the flow family. Like flow matching, they learn a deterministic transport map, which improves training stability and naturally generalizes to multi-step training and generation. Our method can be trained on standard transformer architectures without modification, opening the door to wider adoption.

On ImageNet-256px, our B/2 model approaches the performance of consistency-based XL/2 models because it preserves modeling capacity, while our XL/2 model achieves a new best FID of 2.38 under the same 1NFE setting. Our method also enables fully end-to-end training of 56-layer and 112-layer 1NFE models through depth repetition without any intermediate supervision, achieving FIDs of 2.08 and 1.94 and surpassing their 28-layer 2NFE and 4NFE counterparts.

## 2. Related Works

**The acceleration of flow-based models.** Early distillation works train few-step student models to match the predictions of a teacher flow model (Salimans & Ho, 2022; Liu et al., 2023; Salimans et al., 2024; Yan et al., 2024). Consistency models (CMs) (Song et al., 2023; Song & Dhariwal, 2024) introduce a self-consistency constraint and sup-

port standalone training as a new class of generative models. sCM (Lu & Song, 2025) extends this consistency constraint to continuous time to minimize discretization error. iMM (Zhou et al., 2025) incorporates moment matching. Shortcut (Frans et al., 2025) redefines the boundary condition to allow transport between arbitrary timesteps. Mean-Flow (Geng et al., 2026) and AYF (Sabour et al., 2026) further extend Shortcut to continuous time. However, these methods still tend to produce slightly blurry results on large-scale text-to-image and video tasks (Luo et al., 2023a), so distributional matching methods, such as adversarial training (Lin et al., 2024; Ren et al., 2024; Lin & Yang, 2024; Lin et al., 2025; 2026; Lu et al., 2025; Sauer et al., 2024b;a; Wang et al., 2024; Kohler et al., 2024; Kang et al., 2024; Xu et al., 2024; Chen et al., 2025) and score distillation (Yin et al., 2024b;a; Sauer et al., 2024b; Lu et al., 2025; Luo et al., 2023b; Zheng et al., 2026b), are often incorporated in practice.

**Generative adversarial networks.** Early GAN research developed many techniques that succeeded on domain-specific datasets (Reed et al., 2016; Zhang et al., 2017; Karras et al., 2018; 2019; 2020b; 2021; 2020a). BigGAN (Brock et al., 2019) and StyleGAN-XL (Sauer et al., 2022) further scaled GANs to ImageNet (Russakovsky et al., 2015). However, GANs have fallen out of favor because of their training instability and limited scalability. Several works on large-scale text-to-image generation with GANs still employ convolutional architectures with complex designs (Kang et al., 2023; Zhu et al., 2025). GANs with transformer architectures have been challenging to scale (Jiang et al., 2021; Lee et al., 2022; Hudson & Zitnick, 2021). More recently, R3GAN (Huang et al., 2024) simplifies the adversarial formulation and achieves state-of-the-art performance on the ImageNet-64 benchmark using a convolutional architecture. GAT (Hyun et al., 2025) further extends this line of work to a latent transformer architecture. These works have revitalized interest in adversarial training. However, GAT still employs a non-standard transformer architecture and relies on a pre-trained feature network. Our work combines adversarial models with flow models and improves the training stability of adversarial methods.

## 3. Method

### 3.1. Adversarial Training Preliminaries

Our method builds on generative adversarial networks (GANs), in which a generator $G : \mathbb{R}^m \to \mathbb{R}^n$ aims to transport samples $z$ from a prior distribution $\mathcal{Z}$, *e.g.*, a Gaussian, to samples $x$ from the data distribution $\mathcal{X}$, while a discriminator $D : \mathbb{R}^n \to \mathbb{R}$ aims to distinguish real samples from generated ones. The adversarial optimization involves a minimax game in which $D$ is trained to maximize this

distinction, while $G$ is trained to minimize it:

$$\mathcal{L}_{\text{adv}}^D = \mathbb{E}_{z,x} \left[ f(D(x), D(G(z))) \right], \tag{1}$$

$$\mathcal{L}_{\text{adv}}^G = \mathbb{E}_{z,x} \left[ f(D(G(z)), D(x)) \right]. \tag{2}$$

We adopt the relativistic objective (Jolicoeur-Martineau, 2019), where $f(a, b) = -\log(\text{sigmoid}(a - b))$, because it yields a better loss landscape (Sun et al., 2020) and achieves the current state of the art (Huang et al., 2024; Hyun et al., 2025).

Additionally, gradient penalties $R_1$ and $R_2$ (Roth et al., 2017) are applied to $D$. They prevent $G$ from being pushed away from equilibrium (Mescheder et al., 2018) and impose a constraint on the Lipschitz constant of $D$ (Gulrajani et al., 2017). Directly computing these gradient penalties requires expensive double backpropagation and second-order differentiation, so we use a finite-difference approximation (Lin et al., 2025):

$$\mathcal{L}_{\text{r1}}^D = \mathbb{E}_x \left[ \|\nabla_x D(x)\|_2^2 \right] \tag{3}$$

$$\approx \mathbb{E}_x \left[ \frac{1}{\epsilon^2} \left\| D(x) - D(\mathcal{N}(x, \epsilon^2 \mathbf{I})) \right\|_2^2 \right], \tag{4}$$

$$\mathcal{L}_{\text{r2}}^D = \mathbb{E}_z \left[ \|\nabla_{G(z)} D(G(z))\|_2^2 \right] \tag{5}$$

$$\approx \mathbb{E}_z \left[ \frac{1}{\epsilon^2} \left\| D(G(z)) - D(\mathcal{N}(G(z), \epsilon^2 \mathbf{I})) \right\|_2^2 \right], \tag{6}$$

where $\epsilon$ is set to 0.01. We compute the penalties on only 25% of the samples in each batch and observe no performance degradation.

To prevent the discriminator logits from drifting unboundedly in the relativistic setting, we add a logit-centering penalty, following prior work (Karras et al., 2018):

$$\mathcal{L}_{\text{cp}}^D = \mathbb{E}_{z,x} \left[ (D(x) + D(G(z)))^2 \right]. \tag{7}$$

The final GAN objectives are:

$$\mathcal{L}_{\text{GAN}}^D = \mathcal{L}_{\text{adv}}^D + \lambda_{\text{gp}} \mathcal{L}_{\text{r1}}^D + \lambda_{\text{gp}} \mathcal{L}_{\text{r2}}^D + \lambda_{\text{cp}} \mathcal{L}_{\text{cp}}^D, \tag{8}$$

$$\mathcal{L}_{\text{GAN}}^G = \mathcal{L}_{\text{adv}}^G, \tag{9}$$

where $\lambda_{\text{gp}}$ is a tuned hyperparameter that controls the scale of both gradient penalties, $R_1$ and $R_2$, and $\lambda_{\text{cp}}$ is fixed at 0.01.

The expectations are estimated with Monte Carlo approximations over minibatches during training. The generator $G$ and discriminator $D$ are updated alternately. For conditional generation, the condition $c$ is provided to both networks as $G(z, c)$ and $D(x, c)$. we omit this notation when it is not needed.

### 3.2. Single-step Adversarial Flow Models

The GAN objective above only enforces that $G(z)$ matches the data distribution. However, there are many valid transport maps from $z$ to $x$, and the model is free to learn any

one of them. Our proposed adversarial flow models instead learn a deterministic optimal transport map. This prevents generator drift and stabilizes training.

Formally, in optimal transport theory, Brenier's theorem guarantees the existence of a unique optimal transport map when the source distribution is absolutely continuous, *e.g.*, Gaussian, and the cost function is quadratic.

Accordingly, in adversarial flow models, we parameterize the transport map with a deterministic neural network $G(z)$. We further restrict the prior distribution to have the same dimensionality $n$ as the data distribution, *i.e.*, $x, z \in \mathbb{R}^n$ and $G : \mathbb{R}^n \to \mathbb{R}^n$. This constraint is commonly required by flow-based models and does not reduce the generality of our method.

Our goal is to find an optimal transport map $G^*$ that is both a valid transport map and the map that minimizes the total transport cost under the quadratic cost function $\mathbf{c}(x, z) = \|x - z\|_2^2$. This corresponds to Wasserstein-2 optimal transport:

$$G^* = \arg\min_G \int_{\mathcal{Z}} \mathbf{c}(G(z), z) \, dz. \tag{10}$$

Since the adversarial objective encourages matching between the generated and target distributions, and achieves exact marginal matching at the global optimum under standard assumptions, *e.g.*, infinite capacity and perfect optimization, the validity of the transport map is enforced. Under these assumptions, we find that an optimal transport regularization loss can be applied to $G$ to bias the solution toward the unique optimal transport map $G^*$. This loss minimizes the expectation of $\mathbf{c}(G(z), z)$ over $z$, which by definition minimizes the total transport cost:

$$\boxed{\mathcal{L}_{\text{ot}}^G = \mathbb{E}_z \left[ \frac{1}{n} \|G(z) - z\|_2^2 \right]} \tag{11}$$

$$= \frac{1}{n} \int_{\mathcal{Z}} \|G(z) - z\|_2^2 \, dz. \tag{12}$$

The objectives of adversarial flow models (AF) become:

$$\mathcal{L}_{\text{AF}}^D = \mathcal{L}_{\text{adv}}^D + \lambda_{\text{gp}} \mathcal{L}_{\text{r1}}^D + \lambda_{\text{gp}} \mathcal{L}_{\text{r2}}^D + \lambda_{\text{cp}} \mathcal{L}_{\text{cp}}^D, \tag{13}$$

$$\mathcal{L}_{\text{AF}}^G = \mathcal{L}_{\text{adv}}^G + \boxed{\lambda_{\text{ot}} \mathcal{L}_{\text{ot}}^G}. \tag{14}$$

In practice, however, the validity of the transport map is not strictly enforced, because $\mathcal{L}_{\text{ot}}^G$ competes with $\mathcal{L}_{\text{adv}}^G$ and biases the solution toward $G(z) = z$. We therefore introduce a scaling factor $\lambda_{\text{ot}}$. As illustrated in Figure 2, if this value is too small, optimization may get stuck in local minima; if it is too large, it pushes the model toward the identity map and hurts distribution matching. We therefore adopt a schedule that decreases $\lambda_{\text{ot}}$ over the course of training.

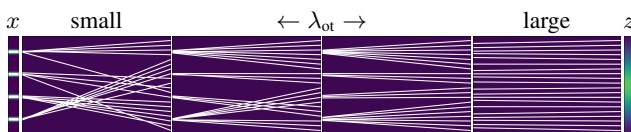

$x$    small    $\leftarrow \lambda_{\text{ot}} \rightarrow$    large    $z$

*Figure 2.* The effect of using different $\lambda_{\text{ot}}$ scales. $\lambda_{\text{ot}} = 0$ is equivalent to GANs. $\lambda_{\text{ot}}$ being too small fails to escape local minima. $\lambda_{\text{ot}}$ being too large forces identity output.

Empirically, we observe that our method consistently learns the same deterministic mapping across different random initializations in the 1D mixture-of-Gaussians experiment (Figure 1), whereas standard GANs produce different transport mappings.

Unlike consistency-based methods, our model does not need to be trained on other timesteps of the probability flow and can instead be trained directly for one-step generation. This saves model capacity, reduces the number of training iterations, and avoids error propagation. Furthermore, in the one-step setting, this formulation removes the hyperparameters associated with timestep sampling and weighting. Our one-step model also avoids teacher forcing entirely.

### 3.3. Multi-step Adversarial Flow Models

Adversarial flow models can be generalized to multi-step generation, allowing the model to transport between arbitrary timesteps along the probability flow. We introduce an interpolation function of the same form used in flow-matching models:

$$x_t = \mathit{interp}(x, z, t) := A(t)x + B(t)z, \qquad (15)$$

where $t \in [0, 1]$. For simplicity, we adopt linear interpolation, with $A(t) = 1 - t$ and $B(t) = t$.

We modify the generator to accept an additional source timestep $s$ and target timestep $t$, yielding $G(x_s, s, t)$, and we modify the discriminator to accept the target timestep, yielding $D(x_t, t)$. During training, $x_s$ and $x_t$ are obtained by interpolating independently sampled $x$ and $z$. The adversarial loss then becomes:

$$\mathcal{L}_{\text{adv}}^D = \mathbb{E}_{x,z,s,t}\left[f(D(x_t, t), D(G(x_s, s, t), t))\right], \qquad (16)$$

$$\mathcal{L}_{\text{adv}}^G = \mathbb{E}_{x,z,s,t}\left[f(D(G(x_s, s, t), t), D(x_t, t))\right]. \qquad (17)$$

The $R_1$ and $R_2$ gradient penalties are modified accordingly. We omit their approximate forms for simplicity:

$$\mathcal{L}_{\text{r1}}^D = \mathbb{E}_{x,z,s,t}\left[w(s, t)\,\|\nabla_{x_t} D(x_t, t)\|_2^2\right], \qquad (18)$$

$$\mathcal{L}_{\text{r2}}^D = \mathbb{E}_{x,z,s,t}\left[w(s, t)\,\|\nabla_{G(x_s, s, t)} D(G(x_s, s, t), t)\|_2^2\right]. \qquad (19)$$

In the multi-step setting, Brenier's theorem still applies because the source distribution remains absolutely continuous, as the interpolation process is equivalent to convolution with a Gaussian. The quadratic optimal transport loss can

be generalized as follows:

$$\mathcal{L}_{\text{ot}}^G = \mathbb{E}_{x,z,s,t}\left[\frac{1}{n} \cdot \frac{1}{w(s,t)} \cdot \|G(x_s, s, t) - x_s\|_2^2\right]. \qquad (20)$$

We empirically find that the following weighting function works well:

$$w(s, t) = \max\left(|s - t|, \delta\right), \qquad (21)$$

where $\delta = 0.001$ is chosen for numerical stability.

During training, the timesteps can be sampled as $s \sim \mathcal{U}(0, 1)$ and $t \sim \mathcal{U}(0, s)$. In this case, the model is trained to support transport between arbitrary timesteps with $t < s$ along the probability flow at generation time. When $s$ and $t$ are close, the model behaves like a flow-matching model. When they are far apart, the model behaves like a trajectory model. Since $G$ directly learns the target distribution through $D$ without requiring consistency propagation, the model can alternatively be trained only on the discrete set of timesteps needed for a specific few-step inference setting, of which single-step generation is a special case. This saves model capacity and training iterations. Our framework therefore extends adversarial training from single-step generation to discrete-time flow modeling.

### 3.4. Discriminator Formulation

It is important **not** to condition $D$ on the source sample. Specifically, formulating the discriminator as $D(x, \underline{z})$ for the single-step setting, or as $D(x_t, t, \underline{x_s, s})$ for the multi-step setting, is incorrect. This is because, during training, $x$ and $z$ are sampled independently. This formulation incorrectly tells $D$ that $z$ should be paired with every $x$, but $G$ can produce only a single mapping. Since this objective is impossible to satisfy, training will oscillate or diverge.

When searching for hyperparameters, one complication we encounter is gradient magnitude. Specifically, the objective for $G$ consists of both the adversarial loss and the optimal transport loss. However, the adversarial gradient received from $D$ can have varying magnitudes, influenced by the architecture, the weight initialization, and the gradient-penalty strength. This makes it difficult to find a value of $\lambda_{\text{ot}}$ that works across model sizes.

Formally, we decompose the gradient of the generator loss with respect to the generator parameter $\theta$ using the chain rule:

$$\begin{aligned}
\frac{\partial \mathcal{L}_{\text{AF}}^G}{\partial \theta} &= \frac{\partial \mathcal{L}_{\text{adv}}^G}{\partial \theta} + \frac{\lambda_{\text{ot}} \partial \mathcal{L}_{\text{ot}}^G}{\partial \theta} \\
&= \frac{\partial G(z)}{\partial \theta} \cdot \boxed{\underbrace{\frac{\partial D(G(z))}{\partial G(z)}}_{\text{discriminator}} \cdot \underbrace{\frac{\partial \mathcal{L}_{\text{adv}}^G}{\partial D(G(z))}}_{\text{adversarial obj.}}} \\
&\quad + \frac{\partial G(z)}{\partial \theta} \cdot \underbrace{\frac{\lambda_{\text{ot}} \partial \mathcal{L}_{\text{ot}}^G}{\partial G(z)}}_{\text{transport obj.}}.
\end{aligned} \qquad (22)$$

The boxed term is the gradient passed down from $D$, and its magnitude can vary substantially. In traditional GANs, only the adversarial loss is used, and adaptive optimizers rescale the magnitude of each parameter update, making $G$ largely invariant to the absolute gradient scale. In our case, however, the magnitude matters because it determines the ratio between the adversarial and optimal transport losses.

We therefore propose a gradient-normalization technique. Specifically, we change the formulation to $D(\phi(G(z)))$, where $\phi$ is the identity operator in the forward pass but rescales the gradient in the backward pass:

$$\phi' = \frac{\frac{\partial \mathcal{L}_{\text{adv}}^G}{\partial G(z)}}{\sqrt{n} \cdot \sqrt{\text{EMA}(\|\frac{\partial \mathcal{L}_{\text{adv}}^G}{\partial G(z)}\|_2^2, \beta_2)}}. \quad (23)$$

The operator $\phi$ tracks the exponential moving average (EMA) of the gradient norm, normalizes the gradient by this average norm, and rescales it by $\frac{1}{\sqrt{n}}$, where $n$ is the data dimensionality. It can be viewed as an extension of Adam (Kingma & Ba, 2015) to the backward path, so we use the same $\beta_2$ as in Adam for the EMA decay. After normalizing the adversarial gradient to a unit scale, we can find a value of $\lambda_{\text{ot}}$ that works well across model sizes.

### 3.5. Connections to Guidance

Control over sampling temperature and conditional alignment is a desirable property. We show that guidance can be incorporated into adversarial flow models. As illustrative examples, we consider classifier guidance (CG) (Dhariwal & Nichol, 2021) and classifier-free guidance (CFG) (Ho & Salimans, 2021), given their popularity in conditional generation.

We visualize an example conditional flow in Figure 3a and the effect of CFG on flow-matching models in Figure 3b. Prior adversarial works (Sauer et al., 2022; Kang et al., 2023) introduce a classifier $C(x, c)$ that predicts $p(c|x)$ and train $G$ with an additional loss that maximizes the classification probability:

$$\mathcal{L}_{\text{cg}}^G = \mathbb{E}_{z,c} \left[ -C(G(z, c), c) \right], \quad (24)$$

$$\mathcal{L}_{\text{AF}}^G = \mathcal{L}_{\text{adv}}^G + \lambda_{\text{ot}} \mathcal{L}_{\text{ot}}^G + \boxed{\lambda_{\text{cg}} \mathcal{L}_{\text{cg}}^G}. \quad (25)$$

However, as Figure 3c shows, the guided model yields results that are almost identical to those of the original model. This is because, in this particular example, the classes are well separated, so the classifier has a clear decision boundary and yields no gradient. This example illustrates an important difference from CFG, whose transport is influenced by guidance gradients accumulated along the flow rather than at a single timestep. The classifier gradients exist at higher timesteps because interpolation diffuses the class boundaries.

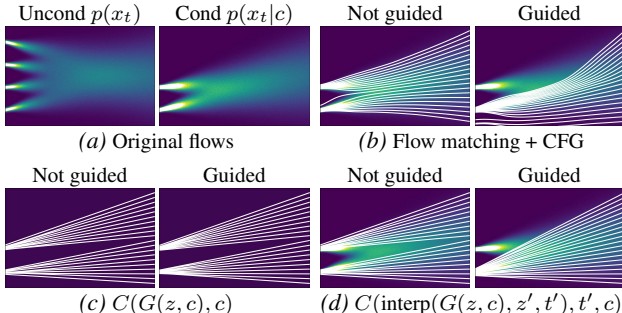

*(a) Original flows*     *(b) Flow matching + CFG*

*(c) $C(G(z, c), c)$*     *(d) $C(interp(G(z, c), z', t'), t', c)$*

*Figure 3.* Conditional models trained on a 1D Gaussian mixture. (a) The conditional flow. (b) The effect of CFG on flow-matching models. (c) Adversarial flow models with simple classifier guidance does not work. (d) Adversarial flow models with flow-based classifier guidance match the behavior of CFG.

To obtain the accumulated guidance gradient along the flow, even for single-step training and generation, we switch to a time-conditioned classifier $C(x_{t'}, t', c)$ that predicts $p(c|x_{t'})$ on the probability flow. During training, $G$ and $D$ are still trained for a single step only. The generated samples $G(z, c)$ are interpolated to random timesteps $t' \sim \mathcal{U}(0, 1)$ with independent noise samples $z' \sim \mathcal{Z}$ before being fed to the time-conditioned classifier:

$$\boxed{\mathcal{L}_{\text{cg}}^G = \mathbb{E}_{z,c,z',t'} \left[ -C(interp(G(z, c), z', t'), t', c) \right].} \quad (26)$$

This maximizes the expected classification score over all timesteps, giving results similar to CFG, as shown in Figure 3d. The timestep $t'$ can be sampled from a custom range, which corresponds to performing CFG on selected timesteps. Equation (24) is a special case of Equation (26) in which $t'$ is always 0. The hyperparameters, *i.e.*, the scale $\lambda_{\text{cg}}$ and the range of $t'$, can optionally be amortized into $G$ to allow inference-time adjustment.

For clarity, in our experiment, $C$ is trained offline as an independent network. If adversarial flow is used to post-train an existing flow-matching model $v$, the gradient of an implicit classifier can be derived from $v$ following CFG:

$$\nabla_{x_t} C(x_t, t, c) = v(x_t, t) - v(x_t, t, c), \quad (27)$$

$$\boxed{\mathcal{L}_{\text{cfg}}^G = \mathbb{E}_{z,c,z',t'} \left[ -\frac{1}{n} G(z, c)^\top \nabla_{(\cdot)} C(\cdot, t', c) \right],} \quad (28)$$

where $(\cdot)$ is shorthand for $interp(G(z, c), z', t')$. Details and derivations are provided in Appendix F.

### 3.6. Different Model Generalization

In theory, under infinite capacity and perfect optimization, flow matching converges to the ground-truth probability flow, and adversarial training reaches equilibrium. Both types of models transport samples to the empirical distribution of the training data and therefore overfit by reproducing only the training samples. In practice, however,

finite-capacity models learn a generalized distribution. Adversarial training can induce a different generalized distribution.

Specifically, flow matching's squared $L_2$ criterion measures isotropic Euclidean distance rather than semantic distance on the data manifold. Furthermore, flow matching, connected to DDPM (Ho et al., 2020), minimizes forward KL divergence to maximize mode coverage. These properties lead to frequent generation of out-of-distribution samples in the guidance-free setting. In contrast, adversarial training uses a discriminator network as a learned criterion. Prior work suggests that deep networks can better capture manifold structure and serve as a better perceptual metric than Euclidean distance (Zhang et al., 2018). Our choice of adversarial objective is also closer to JS divergence (Goodfellow et al., 2014) and is less sensitive to outliers. We hypothesize that these factors help explain why our models even outperform flow matching in the guidance-free setting.

### 3.7. Model Architecture

We parameterize the single-step or multi-step generator $G$ with a neural network $g$. We find that it works equally well when formulated either directly:

$$G(z) = g(z), \tag{29}$$
$$G(x_s, s, t) = g(x_s, s, t), \tag{30}$$

or in residual form:

$$G(z) = z - g(z), \tag{31}$$
$$G(x_s, s, t) = x_s - (s - t)\, g(x_s, s, t). \tag{32}$$

The latter is closely related to the velocity-prediction formulation used in existing flow-matching and consistency-based models. To demonstrate the feasibility of both, we train our single-step models using the direct formulation and our multi-step models using the residual formulation. We parameterize $D$ directly with a neural network $d$.

Both $g$ and $d$ use the standard diffusion transformer (DiT) architecture (Peebles & Xie, 2023). For single-step models, the timestep projection is removed. For multi-step models with fixed discretizations, a single timestep projection is used. For any-step models, two timestep projections are used in $g$. The condition is injected through modulation in both $g$ and $d$, following the original DiT. Our discriminator $d$ is nearly identical to $g$, except for the addition of a learnable [CLS] token prepended to the input. The [CLS] token is used to produce the logit through a final LayerNorm (Ba et al., 2016) and a linear projection. Overall, our architecture requires only minimal modifications to the original DiT.

### 3.8. Deep Model Architecture

Prior research (Lin et al., 2025) indicates that effective model depth is critical for capturing the nonlinear transfor-

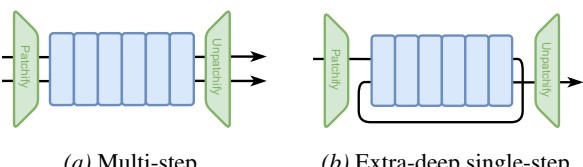

*(a)* Multi-step            *(b)* Extra-deep single-step

*Figure 4.* Deep architecture using transformer block repetition. Extra-deep models are trained end-to-end using single-step objective without any intermediate supervision.

mations required to generate high-quality samples, and that insufficient depth is a primary cause of artifacts in single-step models. We therefore experiment with end-to-end training of extra-deep single-step models. In theory, extra-deep single-step models can outperform their multi-step counterparts because they can pass hidden states end-to-end without projecting into and reinterpreting from the data space, require no manual definition of timestep discretizations, and are trained without teacher forcing.

As illustrated in Figure 4, our extra-deep models use transformer block repetition (Dehghani et al., 2019). The hidden state from the initial pass is recycled. A timestep-like embedding is still provided to the transformer blocks only to distinguish repetition iterations, but the entire network is trained end-to-end for single-forward generation without any intermediate supervision. This design allows us to match the number of parameters and the model behavior of the multi-step counterpart exactly for comparison.

## 4. Experiments

### 4.1. Setups and Training Details

**Experiment setups.** We train on class-conditional ImageNet (Russakovsky et al., 2015) generation to compare with prior work. We follow standard protocols by resizing the images to 256×256 and applying horizontal flips. We use a pre-trained variational autoencoder (VAE)[1] (Rombach et al., 2022) and train the models in $32 \times 32 \times 4$ latent space. Evaluations use Fréchet Inception Distance on 50k class-balanced samples (FID-50k) (Heusel et al., 2017) against the entire train set.

**Training details.** We use an initial learning rate of $1 \times 10^{-4}$ and a batch size of 256 consistent with prior works (Peebles & Xie, 2023; Frans et al., 2025; Geng et al., 2026). We use the AdamW (Loshchilov & Hutter, 2019) optimizer with $\beta_1 = 0, \beta_2 = 0.9$. We set weight decay to 0.01 and EMA decay to 0.9999. We follow MeanFlow's definition of model sizes: B/2, M/2, L/2, XL/2, where 2 denotes the patch size. $G$ and $D$ always use models of the same size. We use separate dataloaders for $G$ and $D$. Epochs are measured by the number of images seen by $G$. Since different models reach their peak FID at different epochs,

---

[1] https://huggingface.co/stabilityai/sd-vae-ft-mse

*Table 1.* **Initial $\lambda_{\text{ot}}$ and $\lambda_{\text{gp}}$.** B/2 model 1NFE. FID-50k measured on 20 epochs. Average of two runs with variation labeled in gray. Without optimal transport loss, training diverges.

| FID↓ | | $\lambda_{\text{ot}}$ | | |
| | 0 | 0.1 | 0.2 | 0.5 |
|---|---|---|---|---|
| | 0.1 | $178.22\pm4$ | $60.20\pm8$ | $70.15\pm8$ | $178.00\pm12$ |
| $\lambda_{\text{gp}}$ 0.25 | $174.93\pm2$ | $54.92\pm2$ | $53.90\pm1$ | $157.06\pm62$ |
| 0.5 | $171.81\pm6$ | $73.85\pm11$ | $57.51\pm1$ | $62.38\pm6$ |

*Table 2.* **Decay of $\lambda_{\text{ot}}$.** B/2 model 1NFE. FID-50k measured on 100 epochs. Decay is critical for achieving peak performance.

| $\lambda_{\text{ot}}$ | Decay | FID↓ |
|---|---|---|
| 0.2 | Constant | 29.4 |
| $0.2 \rightarrow 0.01$ | Cosine decay over 100 epochs | 8.51 |
| $0.2 \rightarrow 0.0$ | Cosine decay over 100 epochs | 8.69 |

we report the earliest epoch at which the best FID is reached.

**Additional training techniques and details.** We provide details of additional training techniques in Appendix C, including the use of EMA reload, discriminator reset, and discriminator augmentation (DA) (Karras et al., 2020a). We provide additional training details in Appendix D, including the learning rate and OT decay.

**Classifier guidance.** We train the model without guidance until it reaches its best FID and then continue training with guidance. Our classifier is trained from scratch on ImageNet using the cross-entropy objective for 30 epochs. It uses the same B/2 transformer architecture in latent space. We do not amortize the scale of CG into models for a fair comparison with prior works.

**Extra-deep models.** We increase depth only in $G$ while keeping $D$ at its standard depth. The learning rate of $G$ is reduced by the repetition factor. Extra-deep models are trained end-to-end from scratch following the single-step objective.

### 4.2. Ablation Studies on the Hyperparameters

**The effect of optimal transport loss.** Table 1 shows a grid search of the optimal initial $\lambda_{ot}$ and $\lambda_{gp}$. Without the OT loss, training diverges regardless of $\lambda_{gp}$, demonstrating the importance of the OT objective for stabilizing adversarial training in transformers. We also observe that an overly small $\lambda_{ot}$ fails to regularize the model, whereas an overly large $\lambda_{ot}$ hurts distribution matching. Table 2 shows that decaying $\lambda_{ot}$ over time is critical for reaching the best FID. In Table 11 of Appendix D, we show that the terminal OT scale can be further lowered given reduced learning rate.

**The effect of flow-based classifier guidance.** Table 3 shows that flow-based CG offers a modest improvement over CG applied only at $t' = 0$. The optimal range, $\mathcal{U}(0, 0.1)$, is much smaller than in typical flow matching, likely because adversarial models already produce good samples without guidance.

*Table 3.* **Classifier guidance scales and timesteps.** XL/2 model 1NFE trained till best FID. Flow-based CG is better.

| FID↓ | | $\lambda_{\text{cg}}$ | |
| | 0.002 | 0.003 | 0.005 |
|---|---|---|---|
| | 0 | 2.48 | 2.40 | 2.47 |
| $t'$ | $\mathcal{U}(0, 0.1)$ | 2.47 | 2.36 | 2.49 |
| | $\mathcal{U}(0, 0.2)$ | 2.52 | 2.42 | 2.48 |
| | $\mathcal{U}(0, 0.5)$ | 2.46 | 2.45 | 2.50 |

### 4.3. Comparisons to the State of the Art

**Single-step with guidance.** Table 4 compares single-step generation with guidance against state-of-the-art consistency-based and GAN models. Under the exact latent space and architectures, denoted by (•), our method achieves new best FIDs with large margins across all model sizes, even when compared to concurrent works (Zhang et al., 2026; Hyun et al., 2025; Wang et al., 2025). Notably, our B/2 model surpasses many XL/2 consistency-based models, likely because B/2 models are capacity-limited and our method does not waste capacity on other timesteps. Our XL/2 model reaches a new best FID of 2.38. Table 8 shows the effect of guidance, and our method with only classifier guidance (CG) and without discriminator augmentation (DA) is still the best. Note that StyleGAN-XL has a slightly better FID while being smaller, but it operates in the pixel space, while ours is restricted by the VAE and DiT patch size 2. We mainly compare our method against others with the same settings denoted by (•).

**Few-step with guidance.** Table 5 shows that our model also achieves better FIDs in the few-step setting. Our models are trained on designated timesteps to preserve capacity. We find that any-step training performs worse due to the dilution of capacity and batch size. This is also observed in our 1D experiments (Figure 1) where adversarial flow models need a larger batch size and converge more slowly for any-step training. In practice, this is often not a limitation, as achieving the best performance in a designated few-step setting is the priority.

**No-guidance generation.** Table 6 shows that, without guidance, even our 1NFE and 2NFE models outperform flow matching with 250NFE+ in the same latent-space setting denoted by (•). This is due to the properties explained in Section 3.6.

**Extra-deep models.** Table 7 shows that our 56-layer and 112-layer models achieve improved FIDs of 2.08 and 1.94, surpassing their 28-layer 2-step and 4-step counterparts. This confirms our hypothesis in Section 3.8. The results reveal an important insight: the quality of single-step generation may not be bounded by the training method, but by the depth of the generator. Depth scaling is therefore a promising direction for future research.

*Table 4.* **Single-step generation on ImageNet 256px.**[1245]

| Method | Param | Epoch | Guidance | NFE | FID↓ |
|---|---|---|---|---|---|
| ***Consistency-based methods*** | | | | | |
| • iCT-XL/2 | 675M | - | None | 1 | 34.24 |
| • Shortcut-XL/2 | 675M | 250 | CFG | 1 | 10.60 |
| • MeanFlow-B/2 | 131M | 240 | CFG | 1 | 6.17 |
| • AlphaFlow-B/2 [†] | 131M | 240 | CFG | 1 | 5.40 |
| • MeanFlow-M/2 | 308M | 240 | CFG | 1 | 5.01 |
| • MeanFlow-L/2 | 459M | 240 | CFG | 1 | 3.84 |
| • MeanFlow-XL/2 | 676M | 240 | CFG | 1 | 3.43 |
| ◦ TiM-XL/2 [†] | 664M | 300 | CFG | 1 | 3.26 |
| • AlphaFlow-XL/2 [†] | 676M | 240 | CFG | 1 | 2.81 |
| ***GANs*** | | | | | |
| BigGAN | 112M | - | cGAN | 1 | 6.95 |
| GigaGAN | 569M | 480 | Match-loss | 1 | 3.45 |
| ◦ GAT-XL/2+REPA [†] | 602M | 40 | DA + cGAN | 1 | 2.96 |
| StyleGAN-XL | 166M | - | CG + cGAN | 1 | 2.30 |
| ***Adversarial flow (Ours)*** | | | | | |
| • AFM-B/2 | 130M | 200 | CG + DA | 1 | 3.05 |
| • AFM-M/2 | 306M | 120 | CG + DA | 1 | 2.82 |
| • AFM-L/2 | 457M | 120 | CG + DA | 1 | 2.63 |
| • AFM-XL/2 | 673M | 125 | CG + DA | 1 | 2.38 |

*Table 5.* **Few-step generation on ImageNet 256px.**[1245]

| Method | Param | Epoch | Guidance | NFE | FID↓ |
|---|---|---|---|---|---|
| ***Consistency-based methods*** | | | | | |
| • iCT-XL/2 | 675M | - | None | 2 | 20.30 |
| • Shortcut-XL/2 | 675M | 250 | CFG | 4 | 7.80 |
| • IMM-XL/2 | 675M | - | CFG | 1×2 | 7.77 |
| • IMM-XL/2 | 675M | - | CFG | 2×2 | 3.99 |
| ◦ TiM-XL/2 [†] | 664M | 300 | CFG | 2×2 | 3.61 |
| • MeanFlow-XL/2 | 676M | 240 | CFG | 2 | 2.93 |
| • MeanFlow-XL/2 | 676M | 1000 | CFG | 2 | 2.20 |
| • AlphaFlow-XL/2 [†] | 676M | 240 | CFG | 2 | 2.16 |
| ***Adversarial flow (Ours)*** | | | | | |
| • AFM-XL/2 | 675M | 95 | CG + DA | 2 | 2.11 |
| • AFM-XL/2 | 675M | 145 | CG + DA | 4 | 2.02 |

*Table 6.* **No-guidance generation on ImageNet 256px.**[1235]

| Method | Param | Epoch | Guidance | NFE | FID↓ |
|---|---|---|---|---|---|
| ***Flow-matching and diffusion*** | | | | | |
| ADM | 554M | 400 | None | 250 | 10.94 |
| • DiT-XL/2 | 675M | 1400 | None | 250 | 9.62 |
| • SiT-XL/2 | 675M | 1400 | None | 250 | 8.30 |
| • SiT-XL/2+Disperse | 675M | 1200 | None | 500 | 7.43 |
| • SiT-XL/2+REPA | 675M | 800 | None | 250 | 5.90 |
| ʀ RAE-XL | 676M | 800 | None | 250 | 1.87 |
| ʀ SiT-XL/2+REPA-E | 675M | 800 | None | 250 | 1.69 |
| ***Autoregressive and masking*** | | | | | |
| MaskGIT | 227M | 300 | None | 8 | 6.18 |
| VAR | 310M | 350 | None | 10 | 4.95 |
| ***Consistency-based methods*** | | | | | |
| • iCT-XL/2 | 675M | - | None | 1 | 34.24 |
| ◦ TiM-XL/2 | 664M | 300 | None | 1 | 7.11 |
| ***Adversarial flow (Ours)*** | | | | | |
| • AFM-B/2 | 130M | 170 | None | 1 | 6.07 |
| • AFM-M/2 | 306M | 110 | None | 1 | 5.21 |
| • AFM-L/2 | 457M | 110 | None | 1 | 4.36 |
| • AFM-XL/2 | 673M | 120 | None | 1 | 3.98 |
| • AFM-XL/2 | 675M | 90 | None | 2 | 2.36 |

*Table 7.* **Deep architectures on ImageNet 256px.**

| Method | Depth | Param | Epoch | Guidance | NFE | FID↓ |
|---|---|---|---|---|---|---|
| AFM-XL/2 | 28 (1×) | 675M | 95 | CG + DA | 2 | 2.11 |
| AFM-XL/2 | 56 (2×) | 675M | 95 | CG + DA | 1 | 2.08 |
| AFM-XL/2 | 28 (1×) | 675M | 145 | CG + DA | 4 | 2.02 |
| AFM-XL/2 | 112 (4×) | 675M | 120 | CG + DA | 1 | 1.94 |

*Table 8.* **Guidance type comparison on ImageNet 256px.**

| Method | Param | Epoch | Guidance | NFE | FID↓ |
|---|---|---|---|---|---|
| AFM-XL/2 | 673M | 120 | None | 1 | 3.98 |
| AFM-XL/2 | 673M | 125 | DA | 1 | 3.86 |
| AFM-XL/2 | 673M | 125 | CG | 1 | 2.54 |
| AFM-XL/2 | 673M | 125 | CG + DA | 1 | 2.38 |

### 4.4. Limitations and Future Works

**Computation efficiency.** Both consistency and adversarial methods require multiple forward passes per iteration, except that adversarial methods commonly use different samples for the independent calculation of the expectations in the $G$ and $D$ losses. Counting epochs by $G$ provides a fair comparison of the number of $G$ updates against consistency methods. We further calculate per-update compute in Appendix E. Our XL/2 1NFE model requires $1.88\times$ the training compute of AlphaFlow but achieves a 15% improvement in best FID. The additional compute comes from the heavy losses and regularization applied to $D$, which could be improved in future work.

**Additional limitations.** (1) $D$ network increases memory consumption. (2) We use CG instead of CFG. (3) Adversarial flow still requires techniques to mitigate the gradient vanishing problem (Appendix C).

**Future works.** (1) The current adversarial flow models are discrete-time flow models. Extending it further to continuous-time flow modeling is a future direction. (2) Our work only explores training from scratch. We leave the exploration of post-training to future work.

## 5. Conclusion

Our work proposes a framework to combine adversarial and flow modeling. We show that learning a deterministic transport greatly stabilizes training. We propose techniques to normalize gradients and incorporate guidance. Our method achieves new best FIDs and demonstrates end-to-end training on 112-layer deep architectures. The framework and findings offer exciting prospects for future research.

---

[1]In tables, (•) use the same latent space and architecture with parameters only differing by the number of timestep embeddings.

[2]In tables, (◦) use the same latent but different architecture.

[3]In tables, (ʀ) use representational latent space.

[4]In tables, (†) are concurrent methods.

[5]In tables, iCT (Song & Dhariwal, 2024) results are reported by iMM (Zhou et al., 2025).

## Acknowledgement

We thank Rohan Choudhury, Chaorui Deng, Peng Wang, and Qing Yan for their valuable discussions on the methodology and manuscript preparation. We thank Yang Zhao and Qi Zhao for their support in developing the dataloading and evaluation infrastructure. We thank Xu Hu for providing access to GPU resources.

## Impact Statement

This paper presents work whose goal is to advance the field of Machine Learning. There are many potential societal consequences of our work, none which we feel must be specifically highlighted here.

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

# A. Quantitative Results

*Table 9.* **Full metrics on conditional ImageNet 256px generation compared with other works.** The table only shows methods and their configurations that are the most comparable to ours for comparison. (•) indicates methods using the same latent space and architecture as ours. Please refer to (Dhariwal & Nichol, 2021) for metric details.

| Method | Space | Param | Epoch | Guidance | NFE | FID↓ | sFID↓ | IS↑ | Prec.↑ | Rec.↑ |
|---|---|---|---|---|---|---|---|---|---|---|
| ***Flow-matching and diffusion*** | | | | | | | | | | |
| ADM | Pixel | 554M | 400 | None | 250 | 10.94 | 6.02 | 100.98 | 0.69 | 0.63 |
| • DiT-XL/2 (Peebles & Xie, 2023) | LDM (Rombach et al., 2022) | 675M | 1400 | None | 250 | 9.62 | 6.85 | 121.50 | 0.67 | 0.67 |
| • SiT-XL/2 (Ma et al., 2024) | LDM (Rombach et al., 2022) | 675M | 1400 | None | 250 | 8.30 | - | - | - | - |
| • SiT-XL/2+Disperse (Wang & He, 2025) | LDM (Rombach et al., 2022) | 675M | 1200 | None | 500 | 7.43 | - | - | - | - |
| • SiT-XL/2+REPA (Yu et al., 2025) | LDM (Rombach et al., 2022) | 675M | 800 | None | 250 | 5.90 | - | - | - | - |
| RAE-XL (Zheng et al., 2026a) | DINOv2 (Oquab et al., 2024) | 676M | 800 | None | 250 | 1.87 | - | 209.70 | 0.80 | 0.63 |
| SiT-XL/2+REPA-E (Leng et al., 2025) | Joint-trained | 675M | 800 | None | 250 | 1.69 | 4.17 | 219.30 | 0.77 | 0.67 |
| ADM-G | Pixel | 554M | 400 | CG | 250×2 | 4.59 | 5.25 | 186.70 | 0.82 | 0.52 |
| • DiT-XL/2 (Peebles & Xie, 2023) | LDM (Rombach et al., 2022) | 675M | 1400 | CFG | 250×2 | 2.27 | 4.60 | 278.24 | 0.83 | 0.57 |
| • SiT-XL/2 (Ma et al., 2024) | LDM (Rombach et al., 2022) | 675M | 1400 | CFG | 250×2 | 2.06 | 4.49 | 277.50 | 0.83 | 0.59 |
| • SiT-XL/2+Disperse (Wang & He, 2025) | LDM (Rombach et al., 2022) | 675M | 1200 | CFG | 500×2 | 1.97 | - | - | - | - |
| • SiT-XL/2+REPA (Yu et al., 2025) | LDM (Rombach et al., 2022) | 675M | 800 | CFG | 250×2 | 1.42 | 4.70 | 305.70 | 0.80 | 0.65 |
| RAE-XL (Zheng et al., 2026a) | DINOv2 (Oquab et al., 2024) | 676M | 800 | AG | 250×2 | 1.41 | - | 309.40 | 0.80 | 0.63 |
| SiT-XL/2+REPA-E (Leng et al., 2025) | Joint-trained | 675M | 800 | CFG | 250×2 | 1.12 | 4.09 | 302.90 | 0.79 | 0.66 |
| ***Consistency-based models*** | | | | | | | | | | |
| • Shortcut-XL/2 (Frans et al., 2025) | | 675M | 250 | CFG | 1 | 10.60 | - | - | - | - |
| TiM-XL/2 (Wang et al., 2025) | | 664M | 300 | None | 1 | 7.11 | 4.97 | 140.39 | 0.71 | 0.63 |
| • MeanFlow-B/2 (Geng et al., 2026) | | 131M | 240 | CFG | 1 | 6.17 | - | - | - | - |
| • AlphaFlow-B/2 (Zhang et al., 2026) | | 131M | 240 | CFG | 1 | 5.40 | - | - | - | - |
| • MeanFlow-M/2 (Geng et al., 2026) | LDM (Rombach et al., 2022) | 308M | 240 | CFG | 1 | 5.01 | - | - | - | - |
| • MeanFlow-L/2 (Geng et al., 2026) | | 459M | 240 | CFG | 1 | 3.84 | - | - | - | - |
| • MeanFlow-XL/2 (Geng et al., 2026) | | 676M | 240 | CFG | 1 | 3.43 | - | - | - | - |
| TiM-XL/2 (Wang et al., 2025) | | 664M | 300 | CFG | 1 | 3.26 | 4.37 | 210.33 | 0.75 | 0.59 |
| • AlphaFlow-XL/2 (Zhang et al., 2026) | | 676M | 240 | CFG | 1 | 2.81 | - | - | - | - |
| • Shortcut-XL/2 (Frans et al., 2025) | | 675M | 250 | CFG | 4 | 7.80 | - | - | - | - |
| • IMM-XL/2 (Zhou et al., 2025) | | 675M | - | CFG | 1×2 | 7.77 | - | - | - | - |
| • IMM-XL/2 (Zhou et al., 2025) | LDM (Rombach et al., 2022) | 675M | - | CFG | 2×2 | 3.99 | - | - | - | - |
| TiM-XL/2 (Wang et al., 2025) | | 664M | 300 | CFG | 2×2 | 3.61 | 6.74 | 151.79 | 0.74 | 0.59 |
| • MeanFlow-XL/2 (Geng et al., 2026) | | 676M | 1000 | CFG | 2 | 2.20 | - | - | - | - |
| • AlphaFlow-XL/2 (Zhang et al., 2026) | | 676M | 240 | CFG | 2 | 2.16 | - | - | - | - |
| ***Autoregressive, masking, and hybrids*** | | | | | | | | | | |
| MaskGIT (Song & Dhariwal, 2024) | VQGAN (Esser et al., 2021) | 227M | 300 | None | 8 | 6.18 | - | 182.10 | 0.80 | 0.51 |
| VAR (Tian et al., 2024) | MS-VQVAE (Tian et al., 2024) | 310M | 350 | None | 10 | 4.95 | - | - | - | - |
| VAR (Tian et al., 2024) | MS-VQVAE (Tian et al., 2024) | 2.0B | 350 | CFG | 10×2 | 1.92 | - | 323.10 | 0.82 | 0.59 |
| MAR (Li et al., 2024) | LDM (Rombach et al., 2022) | 943M | 800 | CFG | 64×100×2 | 1.55 | - | 303.70 | 0.81 | 0.62 |
| ***GANs*** | | | | | | | | | | |
| BigGAN (Brock et al., 2019) | Pixel | 112M | - | cGAN | 1 | 6.95 | 7.36 | 171.40 | 0.87 | 0.28 |
| GigaGAN (Kang et al., 2023) | Pixel | 569M | 480 | Match-loss | 1 | 3.45 | - | 225.52 | 0.84 | 0.61 |
| GAT-XL/2+REPA (Hyun et al., 2025) | LDM (Rombach et al., 2022) | 602M | 40 | DA+cGAN | 1 | 2.96 | - | - | - | - |
| StyleGAN-XL (Sauer et al., 2022) | Pixel | 166M | - | CG+cGAN | 1 | 2.30 | 4.02 | 265.12 | 0.78 | 0.53 |
| ***Adversarial flow models (Ours)*** | | | | | | | | | | |
| • AFM-B/2 | | 130M | 170 | None | 1 | 6.07 | 5.31 | 169.51 | 0.72 | 0.49 |
| • AFM-M/2 | | 306M | 110 | None | 1 | 5.21 | 5.60 | 178.48 | 0.75 | 0.54 |
| • AFM-L/2 | LDM (Rombach et al., 2022) | 457M | 110 | None | 1 | 4.36 | 5.39 | 186.21 | 0.77 | 0.53 |
| • AFM-XL/2 | | 673M | 120 | None | 1 | 3.98 | 5.40 | 201.85 | 0.78 | 0.52 |
| • AFM-XL/2 | | 673M | 90 | None | 2 | 2.36 | 4.35 | 235.77 | 0.81 | 0.52 |
| • AFM-B/2 | | 130M | 200 | CG+DA | 1 | 3.05 | 5.32 | 269.18 | 0.81 | 0.51 |
| • AFM-M/2 | | 306M | 120 | CG+DA | 1 | 2.82 | 5.20 | 279.12 | 0.81 | 0.50 |
| • AFM-L/2 | | 457M | 120 | CG+DA | 1 | 2.63 | 5.10 | 277.96 | 0.81 | 0.52 |
| • AFM-XL/2 | LDM (Rombach et al., 2022) | 673M | 125 | CG+DA | 1 | 2.38 | 4.87 | 284.18 | 0.81 | 0.52 |
| • AFM-XL/2 | | 675M | 95 | CG+DA | 2 | 2.11 | 4.33 | 273.84 | 0.82 | 0.55 |
| • AFM-XL/2 (2×deep, 56-layer) | | 675M | 95 | CG+DA | 1 | 2.08 | 4.79 | 298.33 | 0.79 | 0.56 |
| • AFM-XL/2 | | 675M | 145 | CG+DA | 4 | 2.03 | 4.59 | 259.66 | 0.78 | 0.59 |
| • AFM-XL/2 (4×deep, 112-layer) | | 675M | 120 | CG+DA | 1 | 1.94 | 4.54 | 292.20 | 0.79 | 0.56 |

AG (Karras et al., 2024), CFG (Ho & Salimans, 2021), CG (Dhariwal & Nichol, 2021), cGAN (Miyato & Koyama, 2018), DA (Karras et al., 2020a), Match-loss (Kang et al., 2023)

# B. Qualitative Results

Qualitative results are provided below. Samples are generated with the same seeds across models.

**Deterministic transport.** Deterministic transport behavior is visible, particularly on bald eagle (class 22, first row) and geyser (class 974, last row), where the sky background (blue or white) is usually consistent on the same seed across models. However, details still vary from model to model. This is expected because (1) different model sizes have different degrees of generalization, (2) minibatch training has stochasticity, and (3) the OT scale is reduced toward the end of training.

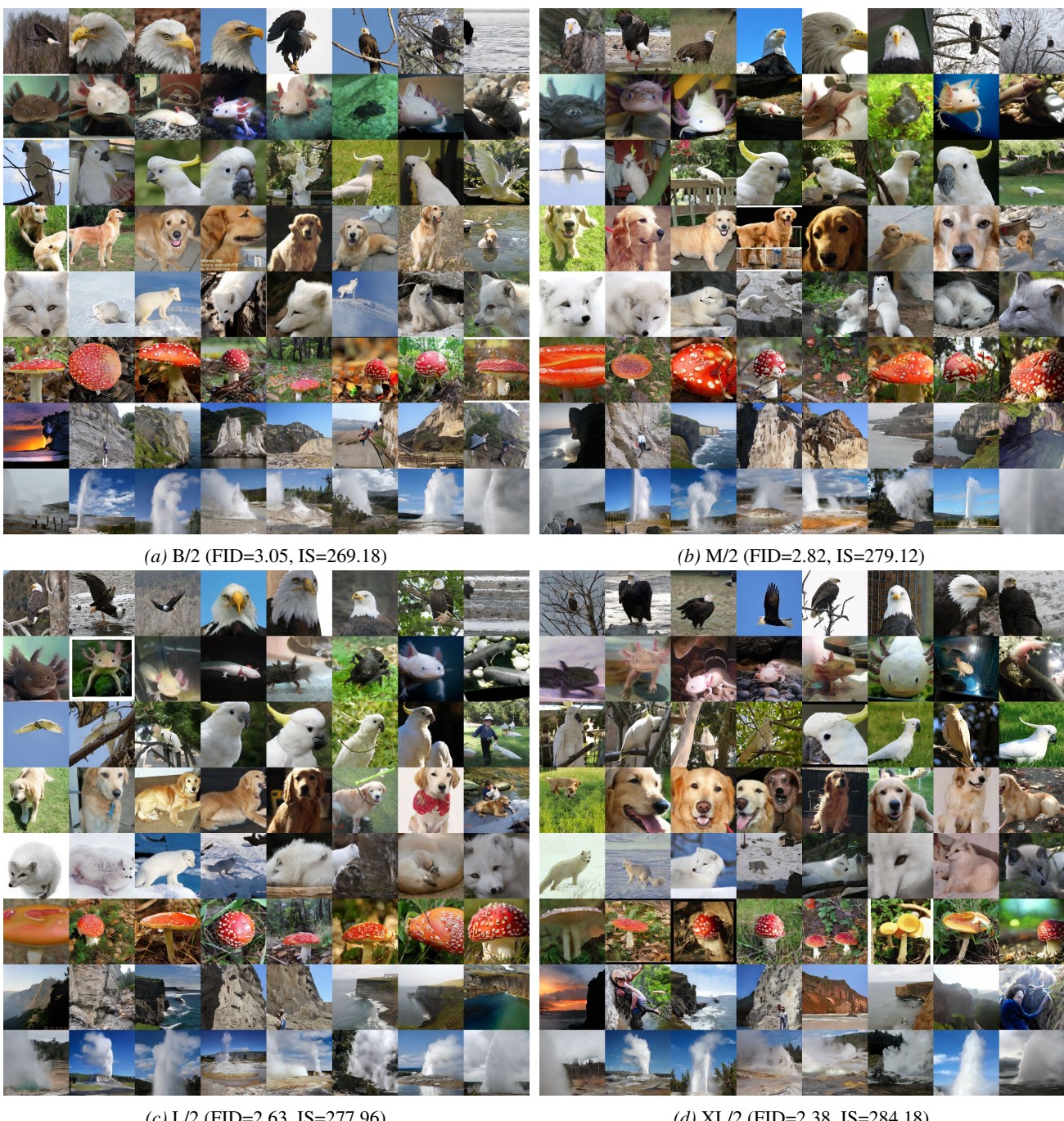

*(a)* B/2 (FID=3.05, IS=269.18)    *(b)* M/2 (FID=2.82, IS=279.12)

*(c)* L/2 (FID=2.63, IS=277.96)    *(d)* XL/2 (FID=2.38, IS=284.18)

*Figure 5.* **Uncurated 1NFE ImageNet-256px generation with guidance.** Classes are (22) bald eagle, (29) axolotl, (89) cockatoo, (207) golden retriever, (279) white fox, (992) agaric, (972) cliff, (974) geyser. Same seeds used across models.

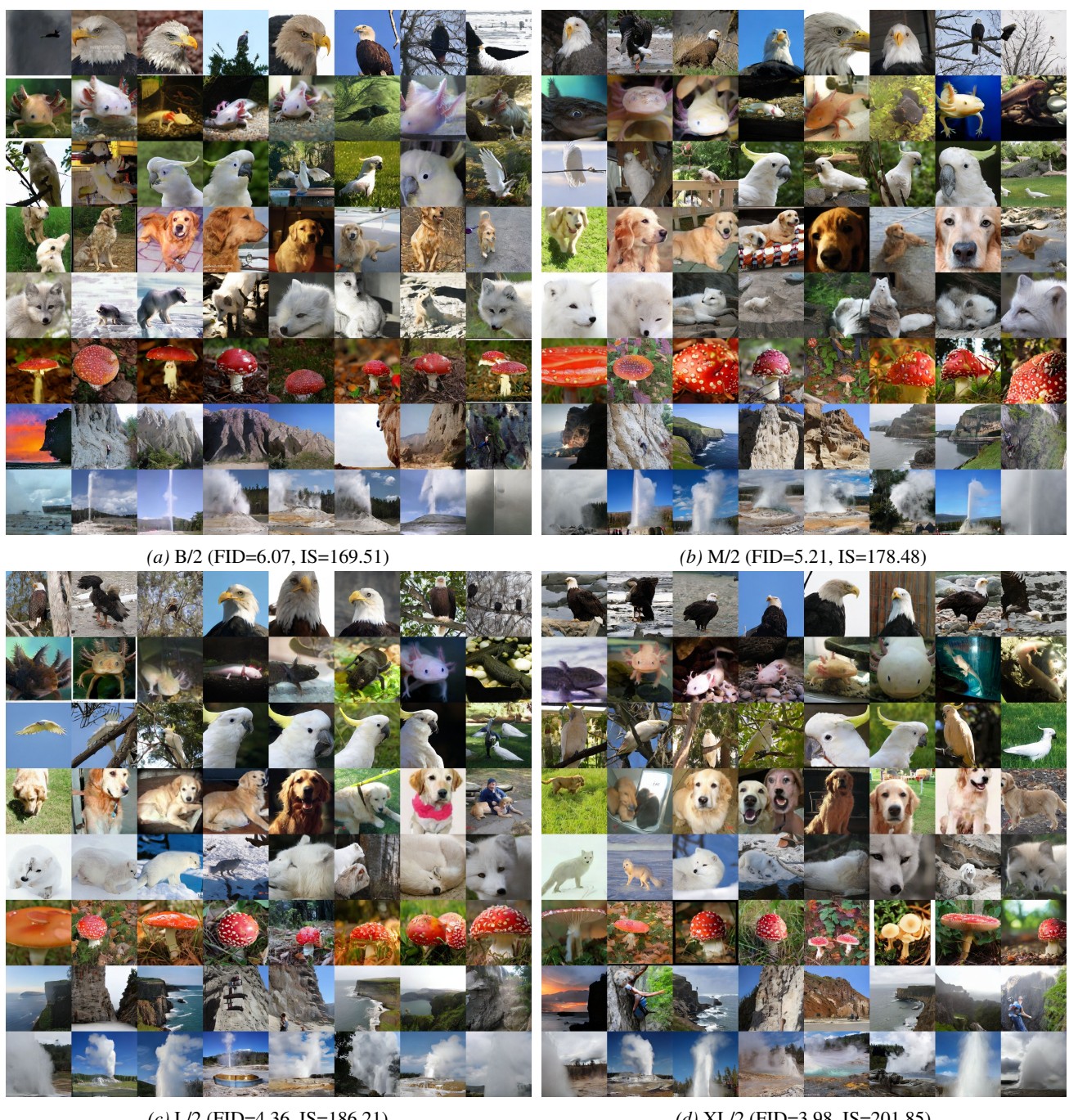

*(a)* B/2 (FID=6.07, IS=169.51)   *(b)* M/2 (FID=5.21, IS=178.48)

*(c)* L/2 (FID=4.36, IS=186.21)   *(d)* XL/2 (FID=3.98, IS=201.85)

*Figure 6.* **Uncurated 1NFE ImageNet-256px generation without guidance.** Classes are (22) bald eagle, (29) axolotl, (89) cockatoo, (207) golden retriever, (279) white fox, (992) agaric, (972) cliff, (974) geyser. Same seeds used across models.

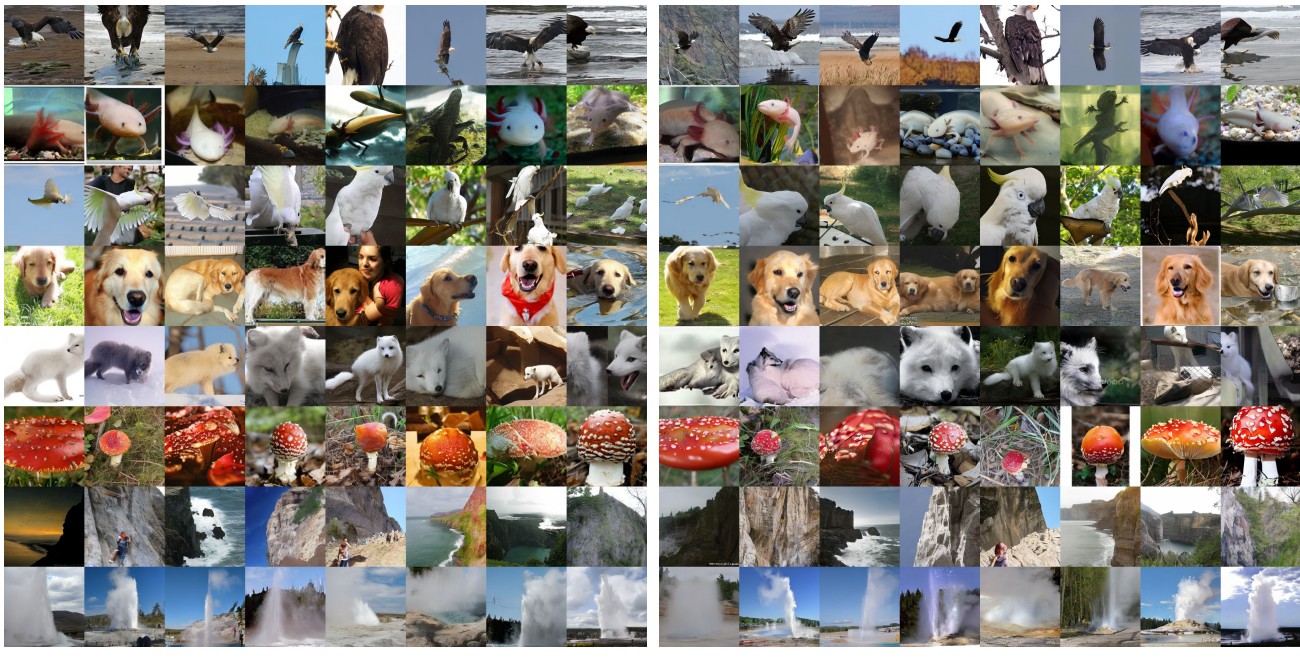

*(a)* XL/2 2NFE (FID=2.11, IS=273.84)    *(b)* XL/2 4NFE (FID=2.03, IS=259.66)

*Figure 7.* **Uncurated 2NFE and 4NFE ImageNet-256px generation with guidance.** Classes are (22) bald eagle, (29) axolotl, (89) cockatoo, (207) golden retriever, (279) white fox, (992) agaric, (972) cliff, (974) geyser. Same seeds used across models.

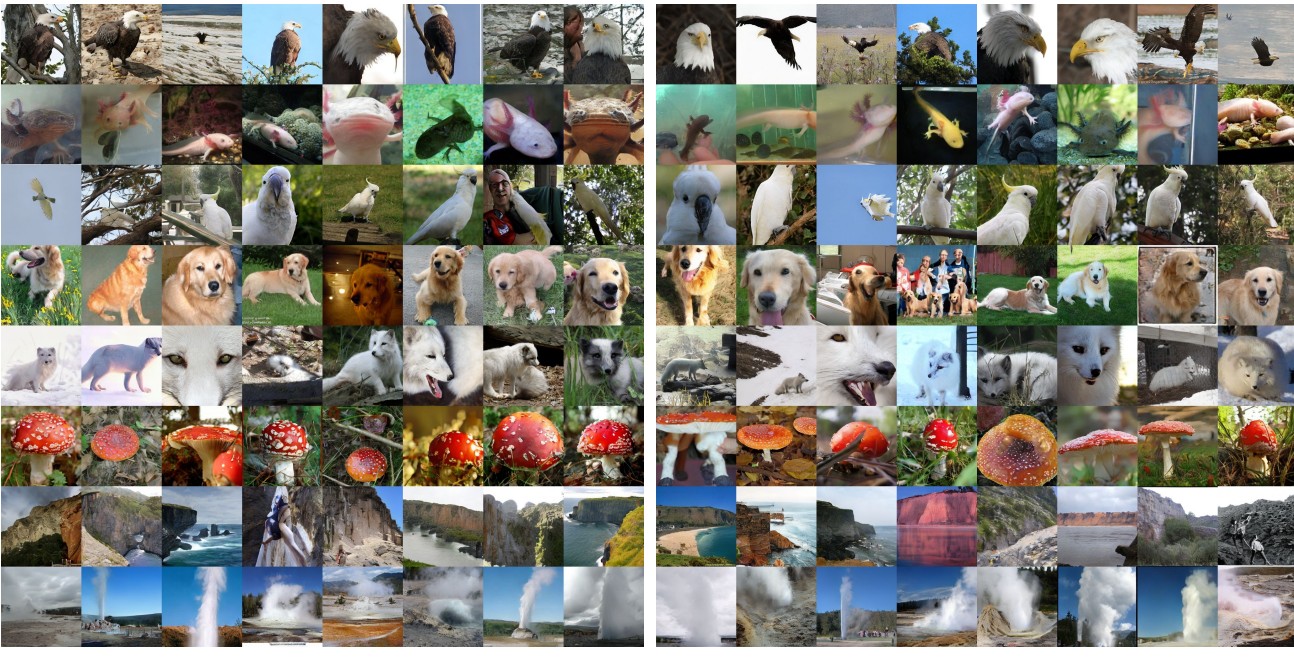

*(a)* XL/2 56-layer 1NFE (FID=2.08, IS=298.33)    *(b)* XL/2 112-layer 1NFE (FID=1.94, IS=292.20)

*Figure 8.* **Uncurated extra-deep 1NFE ImageNet-256px generation with guidance.** Classes are (22) bald eagle, (29) axolotl, (89) cockatoo, (207) golden retriever, (279) white fox, (992) agaric, (972) cliff, (974) geyser. Same seeds used across models.

**Comparisons with flow-matching models.** Figure 9 shows comparisons against SiT (Ma et al., 2024). We show that the adversarial objective generates perceptually better-looking samples even without guidance, while the flow-matching method generates more perceptually out-of-distribution samples without guidance.

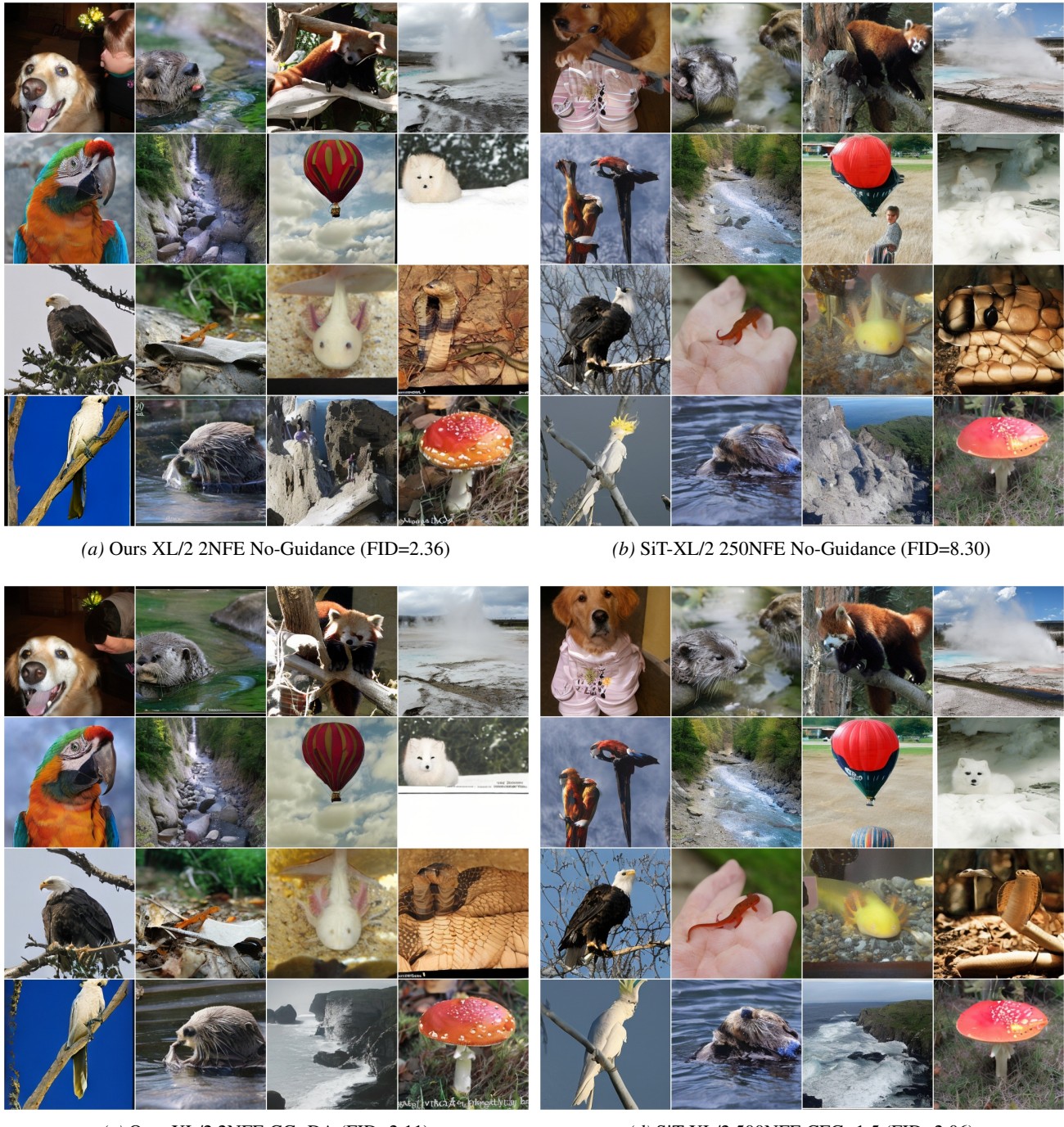

(a) Ours XL/2 2NFE No-Guidance (FID=2.36)      (b) SiT-XL/2 250NFE No-Guidance (FID=8.30)

(c) Ours XL/2 2NFE CG+DA (FID=2.11)      (d) SiT-XL/2 500NFE CFG=1.5 (FID=2.06)

*Figure 9.* **Uncurated comparisons with SiT (Ma et al., 2024).** Same seeds. SiT uses Euler sampler. Classes are (207) golden retriever, (360) otter, (387) red panda, (974) geyser, (88) macaw, (979) valley, (417) balloon, (279) white fox, (22) bald eagle, (27) eft, (29) axolotl, (63) Indian cobra, (89) cockatoo, (360) otter, (972) cliff, (992) agaric.

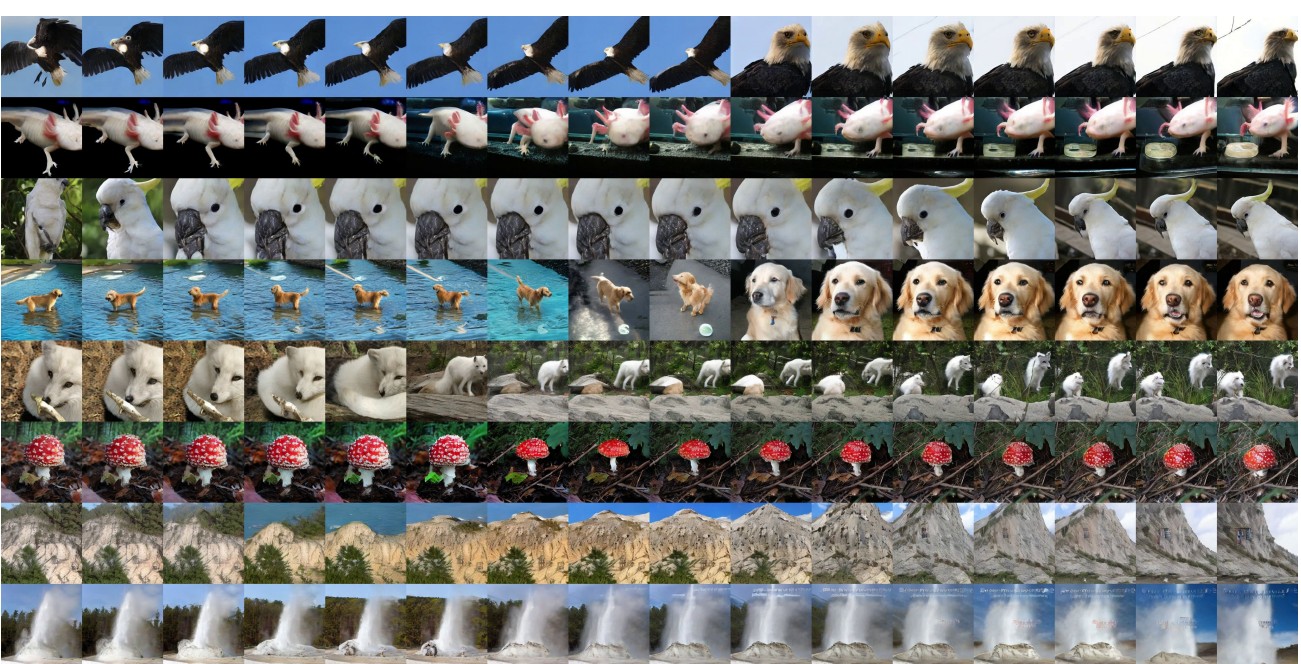

*Figure 10.* **Latent interpolation.** XL/2 1NFE (FID=2.38, IS=284.18). Uncurated.
The figure shows $G(z_t, c)$, where $z_t = \text{slerp}(z_1, z_2, t) := \cos(\frac{\pi}{2}t)z_1 + \sin(\frac{\pi}{2}t)z_2$, $t \in [0, 1]$, $z_1, z_2 \sim \mathcal{N}(0, \mathbf{I})$.

**Layer visualization.** We follow prior works to project the hidden features of every layer onto the top three PCA components (Hyun et al., 2025), and through a trained linear projection and decode them into images using the VAE (Lin et al., 2025). Note that in PCA, because DiT (Peebles & Xie, 2023) uses absolute positional encoding through input, the PCA of some early layers is dominated by sinusoidal encoding. Also, because the PCA of each layer is fitted independently, some layers have a different top-three ordering, so the color of the visualization can change abruptly despite the underlying features being similar. Clear imagery is formed in the later layers of the model. Unlike (Hyun et al., 2025), we do not impose any manual supervision losses on the intermediate features, and the models still obtain top FIDs. Notice that for the 112-layer model, a large number of middle layers seem not to be contributing much in the visualization, but they are indeed effective in improving the final FID. The visualizations may not reveal the full contributions of these layers.

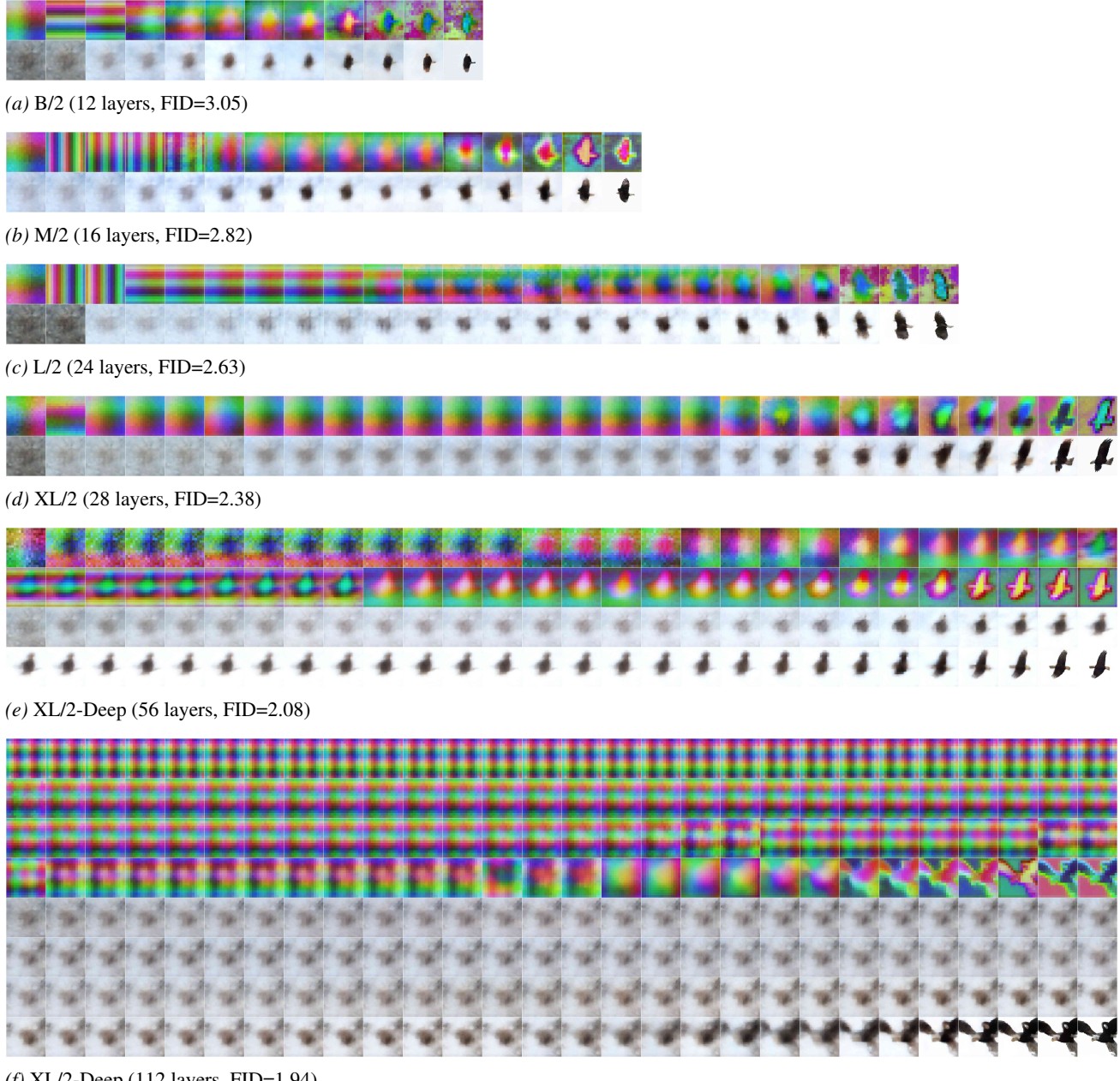

*(a)* B/2 (12 layers, FID=3.05)

*(b)* M/2 (16 layers, FID=2.82)

*(c)* L/2 (24 layers, FID=2.63)

*(d)* XL/2 (28 layers, FID=2.38)

*(e)* XL/2-Deep (56 layers, FID=2.08)

*(f)* XL/2-Deep (112 layers, FID=1.94)

*Figure 11.* **Layer inspection.** Example 1. Class is (22) bald eagle.
The top row is hidden features at every layer projected onto the top-3 PCA components (Hyun et al., 2025).
The bottom row is through a trained linear projection layer and decoded by the VAE (Lin et al., 2025).

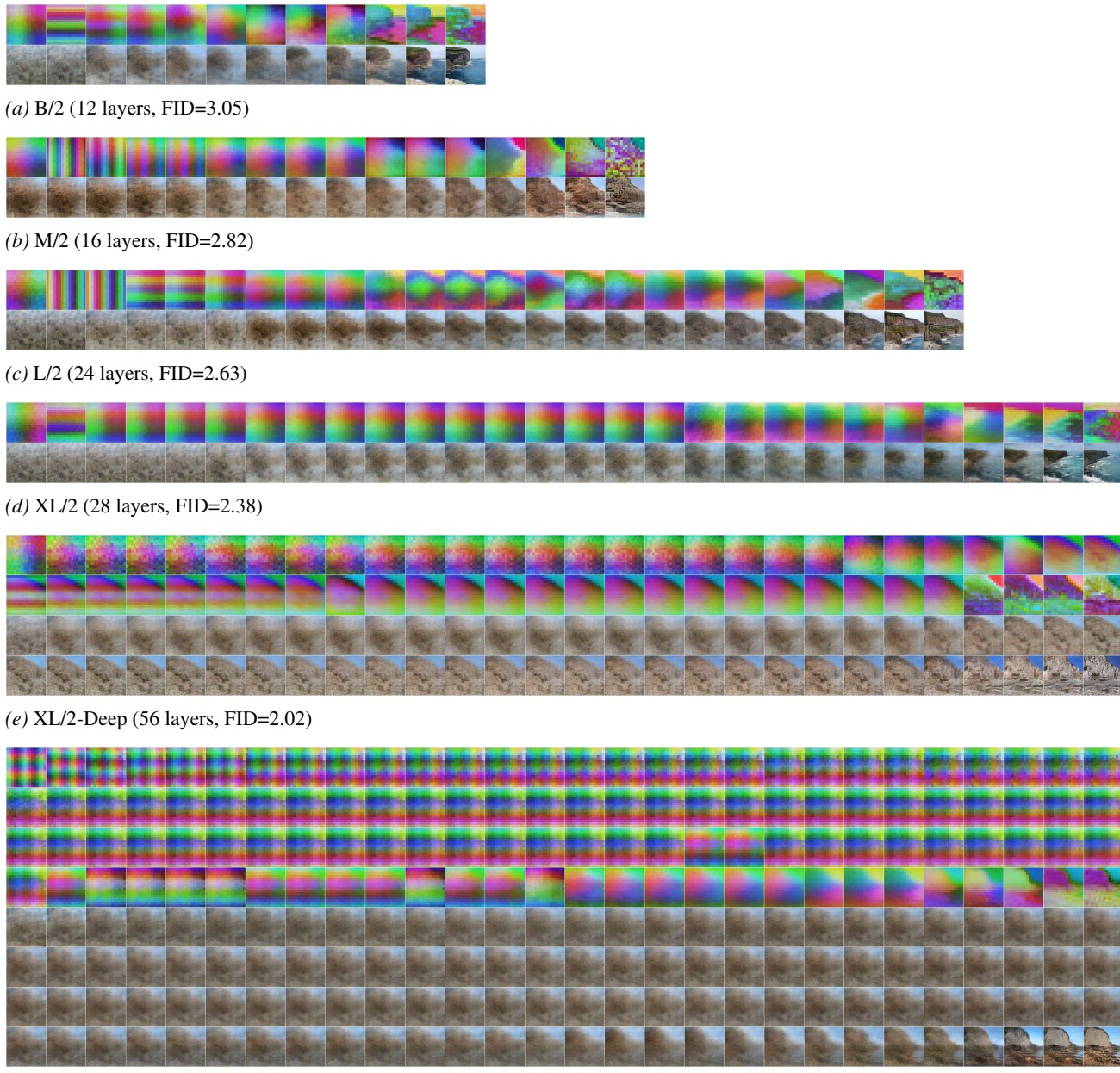

*(a)* B/2 (12 layers, FID=3.05)

*(b)* M/2 (16 layers, FID=2.82)

*(c)* L/2 (24 layers, FID=2.63)

*(d)* XL/2 (28 layers, FID=2.38)

*(e)* XL/2-Deep (56 layers, FID=2.02)

*(f)* XL/2-Deep (112 layers, FID=1.94)

*Figure 12.* **Layer inspection.** Example 2. Class is (972) cliff.
The top row is hidden features at every layer projected onto the top-3 PCA components (Hyun et al., 2025).
The bottom row is through a trained linear projection layer and decoded by the VAE (Lin et al., 2025).

# C. Techniques for Adversarial Training

We share additional training techniques that, while imperfect, are very effective. They address the remaining challenges of adversarial training, not introduced by our adversarial flow models.

On optimization, minimax optimization with gradient descent is prone to oscillation. The equilibrium is better achieved by the weight average than at the last iterate (Daskalakis et al., 2018). We find the optimistic optimizer (Daskalakis et al., 2018) and asymmetrical learning rates (Heusel et al., 2017) ineffective. Our approach is to keep an EMA of $G$ (Yazıcı et al., 2019), which consistently outperforms the online model by a large margin. More importantly, once the performance of the EMA model plateaus toward the end of training, we replace online $G$ with the EMA weights and repeat this procedure automatically after every subsequent epoch. The learning rate is also decreased during this phase. We find this simple technique very effective for approaching peak performance.

On training dynamics, many techniques have been proposed to address the vanishing gradient problem. WGAN (Arjovsky et al., 2017) proposes learning the Wasserstein-1 distance but requires a K-Lipschitz $D$. DiffusionGAN (Wang et al., 2023) projects $D$ onto a flow process in the same spirit as our approach with guidance. Discriminator augmentation (DA) (Karras et al., 2020a) is another approach to increase the support overlap. It has been used by recent GANs (Huang et al., 2024; Hyun et al., 2025). However, we believe that the choice of augmentation may implicitly inject inductive biases. For example, prior work finds that affine transforms outperform other distortions on image data (Karras et al., 2020a) because it implicitly encourages $D$ to recognize affine transforms as a more acceptable generalization for image generation. The last approach is to simply reload $D$ from an earlier checkpoint to reset the pace. Our experiments find that reloading $D$ performs surprisingly well, whereas DiffusionGAN is less effective in our preliminary studies. Therefore, we reload $D$ when training stalls for the no-guidance setting to avoid introducing any additional biases, and additionally use DA for the guidance setting.

Note that our work mainly focuses on stabilizing the generator through learning a deterministic transport map, generalizing adversarial training to flow modeling, and introducing the support for guidance. Our work does not address the gradient vanishing problem and relies on techniques proposed by previous research.

# D. Experimental Details

**Training.** Table 10 provides the details of our architecture and constant hyperparameters. Table 11 lists the training

schedule of our models. The hyperparameter patterns are those that we find during training to produce the best FID. We expect that they could be placed on an automatic schedule, but we leave it to future work. When EMA reload is used, the model is automatically reloaded after every epoch onward. When $D$ reload is used, we manually test a few checkpoints from different epochs and reload only when the training stalls again.

**Precision.** We use TF32 precision to match most prior works. We notice that some prior works (Wang et al., 2025; Hyun et al., 2025) train in BF16, but all adopt modified architectures with the addition of QK-normalization and other changes. We also find that QK-normalization is critical for the training stability in BF16 (Esser et al., 2024). However, we do not see a major throughput improvement on the 256px latent setting with a patch size of 2. Therefore, we stick to the unmodified architecture and TF32.

**Guidance.** Flow-based guidance as discussed in Section 3.5 is used for 1NFE models. For multi-step models, we find it is sufficient to only apply guidance on selected target timesteps as indicated in the table. We find that multi-step models need a slightly higher guidance scale to get the best FID and are adjusted accordingly. For the deep models, we keep the exact setting as any other 1NFE models. When training the classifier, we add affine transforms to data augmentation, and it yields better downstream performance.

*Table 10.* **Architectures and constant hyperparameters.**

| Config | NFE | Network | B/2 | M/2 | L/2 | XL/2 | XL/2-Deep |
|---|---|---|---|---|---|---|---|
| Param | 1 | $G$ | 130M | 306M | 457M | 673M | 675M |
| | | $D$ | 129M | 304M | 455M | 671M | 671M |
| | 2,4 | $G$ | - | - | - | 675M | - |
| | | $D$ | - | - | - | 672M | - |
| Depth | | $G$ | 12 | 16 | 24 | 28 | 28×{2,4} |
| | | $D$ | 12 | 16 | 24 | 28 | 28 |
| Dim | | | 768 | 1024 | 1024 | 1152 | 1152 |
| Heads | | | 12 | 16 | 16 | 16 | 16 |
| Patch size | | | | | 2×2 | | |
| Activation | | | | | GeLU | | |
| MLP expand | | | | | ×4 | | |
| Norm | | | | Pre-LayerNorm + AdaZero | | | |
| Batch size | | | | | 256 | | |
| EMA decay | | | | | 0.9999 | | |
| GP scale $\lambda_{gp}$ | | | | | 0.25 | | |
| GP batch ratio | | | | | 25% | | |
| GP approx. $\epsilon$ | | | | | 0.01 | | |
| Logit-centering $\lambda_{cp}$ | | | | | 0.01 | | |
| AdamW weight decay | | | | | 0.01 | | |
| AdamW betas | | | | | (0.0, 0.9) | | |
| Precision | | | | | TF32 | | |
| Data x-flip | | | | | 0.5 | | |

*Table 11.* **Training schedules and dynamic hyperparameters.**

| | Guidance $\lambda_{\mathrm{cg}}, t'$ | $\lambda_{\mathrm{ot}}$ | LR $G,D$ | EMA Reload | $D$ Reload | Epoch | FID |
|---|---|---|---|---|---|---|---|
| B/2 | None | 0.2→0.01 | 1e-4 | | | 150 | 7.30 |
| 1NFE | None | 0.001 | 8e-5 | Yes | | 155 | 7.05 |
| | None | 0.001 | 3e-5 | Yes | Yes | 170 | 6.07 |
| | 0.003, $\mathcal{U}(0,0.1)$ | 0.001 | 5e-5 | Yes | Yes | 200 | 3.05 |
| M/2 | None | 0.2→0.005 | 1e-4 | | | 100 | 6.19 |
| 1NFE | None | 0.001 | 8e-5 | Yes | | 105 | 5.54 |
| | None | 0.001 | 3e-5 | Yes | Yes | 110 | 5.21 |
| | 0.003, $\mathcal{U}(0,0.1)$ | 0.001 | 5e-5 | Yes | Yes | 120 | 2.82 |
| L/2 | None | 0.2→0.005 | 1e-4 | | | 85 | 6.26 |
| 1NFE | None | 0.001 | 8e-5 | Yes | | 105 | 5.14 |
| | None | 0.001 | 3e-5 | Yes | Yes | 110 | 4.36 |
| | 0.003, $\mathcal{U}(0,0.1)$ | 0.001 | 5e-5 | Yes | Yes | 120 | 2.63 |
| XL/2 | None | 0.2→0.005 | 1e-4 | | | 90 | 5.88 |
| 1NFE | None | 0.001 | 8e-5 | Yes | | 110 | 4.81 |
| | None | 0.001 | 3e-5 | Yes | Yes | 120 | 3.98 |
| | 0.003, $\mathcal{U}(0,0.1)$ | 0.001 | 5e-5 | Yes | Yes | 125 | 2.38 |
| XL/2 | None | 0.25→0.005 | 1e-4 | | | 75 | 4.79 |
| 2NFE | None | 0.001 | 8e-5 | Yes | | 85 | 4.34 |
| | None | 0.001 | 3e-5 | Yes | Yes | 90 | 2.36 |
| | 0.02, [0] | 0.001 | 3e-5 | Yes | Yes | 95 | 2.11 |
| XL/2 | None | 0.25→0.005 | 1e-4 | | | 130 | 5.28 |
| 4NFE | None | 0.001 | 8e-5 | Yes | | 135 | 3.89 |
| | None | 0.001 | 3e-5 | Yes | Yes | 140 | 2.70 |
| | 0.02, [0,0.25] | 0.001 | 3e-5 | Yes | Yes | 145 | 2.02 |
| XL/2 | None | 0.2→0.005 | 5e-5,1e-4 | | | 75 | 4.16 |
| 56L | None | 0.001 | 4e-5,8e-5 | Yes | | 80 | 3.41 |
| 1NFE | None | 0.001 | 1.5e-5,3e-5 | Yes | Yes | 85 | 2.77 |
| | 0.003, $\mathcal{U}(0,0.1)$ | 0.001 | 1e-5,2e-5 | Yes | Yes | 95 | 2.08 |
| XL/2 | None | 0.2→0.005 | 2.5e-5,1e-4 | | | 90 | 3.78 |
| 112L | None | 0.001 | 2e-5,8e-5 | Yes | | 95 | 3.40 |
| 1NFE | None | 0.001 | 2.5e-6,1e-5 | Yes | Yes | 100 | 2.92 |
| | 0.003, $\mathcal{U}(0,0.1)$ | 0.001 | 5e-6,2e-5 | Yes | Yes | 120 | 1.94 |

**Evaluation.** We perform class-balanced evaluation, where we generate 50 images per class for 1000 classes to constitute the total 50k evaluation samples. Recent works (Zhang et al., 2026; Zheng et al., 2026a) find that this approach reduces the stochasticity in the evaluation process and yields more accurate model evaluations. Note that class-balanced evaluation may yield a 0.1 FID advantage compared to class-imbalanced counterparts. However, many prior works do not explicitly state the details of their evaluation protocols. We hence report class-balanced FIDs if a work explicitly provides them, and otherwise report the original metrics from that work. Our method yields a large-margin improvement on FID, especially surpassing the best consistency-based method AlphaFlow (Zhang et al., 2026), which is evaluated in the same class-balanced setting.

The FID and other metrics are computed using the code provided by ADM (Dhariwal & Nichol, 2021). The FID is computed against the entire training set using the precomputed statistics by ADM (Dhariwal & Nichol, 2021).

**Discriminator augmentation.** Discriminator augmentation (DA) is used along with classifier guidance (CG). DA is directly performed in the latent space. Let $\rho$ denote the

augmentation operation, $\phi$ is the gradient normalization operation, the exact form of the final adversarial loss is:

$$\mathcal{L}_{\mathrm{adv}}^{D} = \mathbb{E}_{z,x,\rho}\left[f(D(\phi(\rho(x))), D(\phi(\rho(G(z)))))\right], \quad (33)$$

$$\mathcal{L}_{\mathrm{adv}}^{G} = \mathbb{E}_{z,x,\rho}\left[f(D(\phi(\rho(G(z)))), D(\phi(\rho(x))))\right]. \quad (34)$$

Notice that the same random augmentation $\rho$ must be applied to the real and generated samples as a pair when calculating the expectation of the relativistic loss. We only perform integer translation and cutout with a fixed probability in the latent space. Algorithm 1 shows our implementation using Kornia (Riba et al., 2020). Figure 13 shows the visualization of DA.

---

**Algorithm 1** Discriminator augmentation.

```
1  Sequential(
2      RandomTranslate(
3          p=0.4,
4          translate_x=(0, 0.3),
5          translate_y=(0, 0.3),
6          resample="NEAREST",
7      ),
8      RandomErasing(p=0.4, scale=(0.1, 0.5)),
9      RandomErasing(p=0.4, scale=(0.1, 0.5)),
10     RandomErasing(p=0.4, scale=(0.1, 0.5)),
11 )
```

---

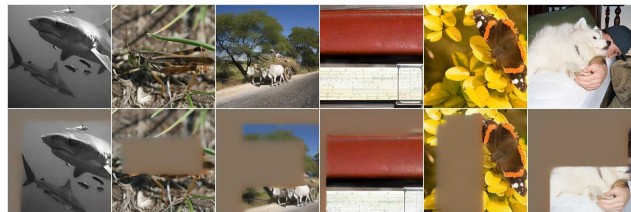

*Figure 13.* **Discriminator augmentation.** The top row is the original samples. The bottom row is what $D$ sees after augmentation. The augmentation is performed in the latent space. The visualization is through the VAE decoder.

**No data leakage.** Prior work (Kynkäänniemi et al., 2023) finds that using a pre-trained ImageNet classifier as $D$ backbone when training GANs on FFHQ (Karras et al., 2019) and LSUN (Yu et al., 2015) generation tasks cheats the FID evaluation by InceptionV3 (Szegedy et al., 2016). Unlike (Huang et al., 2024; Hyun et al., 2025), we do not consider StyleGAN-XL (Sauer et al., 2022) trained on ImageNet fits this description of data leakage. Our method differs further in that it uses the classifier only for guidance. Since CG is an established approach from ADM (Dhariwal & Nichol, 2021) and CFG is just CG with an implicit classifier (Ho & Salimans, 2021), CG should not introduce additional advantages.

# E. Computational Efficiency

As discussed in the main text, both consistency and our methods require multiple forward passes per update iteration. The difference is that consistency-based models use the same data samples across all the forward passes, while adversarial methods have traditionally used different data samples to independently calculate the expectations in the $G$ and $D$ losses. We count the epoch by $G$ as a fair reflection of the number of optimizer update steps taken by $G$. Since our model only needs to be trained on selected timesteps, our $G$ indeed reaches peak performance using fewer optimizer update steps, given the same learning rate and batch size compared to consistency models.

However, when we analyze the computation required per $G$ update step, current adversarial methods are still at a disadvantage due to the excessive computation and regularizations needed by $D$. This is a limitation of the adversarial formulation, not introduced by our adversarial flow.

Table 12 shows the breakdown of the computation per $G$ update. The current adversarial approach consumes $3.625\times$ compute per $G$'s update compared to MeanFlow. After accounting for the reduced number of iterations needed, on XL/2 models, our method uses $1.88\times$ compute compared to MeanFlow but obtains 15% FID improvement. For adversarial methods, it may be possible to reduce the batch size on $D$, apply clever result reuse, explore other methods to limit the Lipschitz constant other than using gradient penalties, and explore other objective functions than the relativistic one to save compute. But for our research, we take the most conservative approach and mostly follow the formulation of prior state-of-the-art GANs (Huang et al., 2024; Hyun et al., 2025). We leave the optimization to future work.

*Table 12.* **Computation analysis.** No guidance. ($^1$) denotes backward pass only compute gradient regarding to input, which consumes $1\times$ compute as forward pass. ($^2$) denotes backward pass also compute gradient regarding to parameters, which consumes $2\times$ compute as forward pass. 2.5=1+1+0.25+0.25, where the 0.25s are the R1 and R2 gradient penalties computed on 25% of samples. Speed is adjusted by best XL/2 epoch: FM: 1400 epochs, MF: 240 epochs, AF: 125 epochs.

| | Generator | | Discriminator | | Total | Speed |
| | Forward | Backward | Forward | Backward | | (Adj. by epoch) |
|---|---|---|---|---|---|---|
| Flow Matching | $G$ | $G^2$ | - | - | 3 | $4.38\times$ |
| MeanFlow | $G + G_{jvp}$ | $G^2$ | - | - | 4 | $1\times$ |
| Adversarial Flow | $G + 2D$ | $G^2 + D^1$ | $G + 2.5D$ | $2.5D^2$ | 14.5 | $1.88\times$ |

# F. Guidance Details

**Derivation of an implicit classifier.** We show the derivation of Equation (28) in the main text. CFG (Ho & Salimans, 2021) shows that an implicit classifier can be derived following Bayes' rule:

$$
\begin{aligned}
\log p(c|x_t) &= \log \frac{p(x_t|c)p(c)}{p(x_t)} \\
&= \log p(x_t|c) + \log p(c) - \log p(x_t) \\
&\sim \log p(x_t|c) - \log p(x_t)
\end{aligned}
\tag{35}
$$

In flow-matching models, the predicted velocity $v(x_t, t) = x_1 - x_0$ is proportional to the negative score $-\nabla_{x_t} \log p(x_t|c)$. Therefore, the gradient of an implicit classifier is derived:

$$
\begin{aligned}
\nabla_{x_t} C(x_t, t, c) &= \nabla_{x_t} \log p(c|x_t) \\
&\sim \nabla_{x_t} \log p(x_t|c) - \nabla_{x_t} \log p(x_t) \\
&\sim (-v(x_t, t, c)) - (-v(x_t, t)) \\
&= v(x_t, t) - v(x_t, t, c)
\end{aligned}
\tag{36}
$$

Given the gradient of an implicit classifier, we want to pass it directly through the generator $G$ in the backpropagation process. This can be achieved using the constant multiple rule trick, where $f(x) = ax$, $f'(x) = a$. So we multiply the gradient of the implicit classifier by the generator output:

$$
\mathcal{L}_{cfg}^G = \mathbb{E}_{z,c,z',t'} \left[ -\frac{1}{n} G(z,c)^\top \nabla_{(\cdot)} C(\cdot, t', c) \right],
\tag{37}
$$

where $(\cdot)$ is short for the flow interpolation process $interp(G(z,c), z', t')$ before passing to the implicit classifier. The negative sign is used to convert loss minimization into classification maximization.

**Multi-step guidance.** The main text only describes our flow-based guidance approach for single-step generation. Since we find that we do not need very strong guidance for ImageNet generation, our multi-step models simply apply guidance by setting $t' = t$:

$$
\mathcal{L}_{cg}^G = \mathbb{E}_{x_s,s,t,c} \left[ -C(G(x_s, s, t, c), t, c) \right].
\tag{38}
$$

**Implicit guidance in prior GAN methods.** Many techniques used in prior GAN works are, in fact, classifier guidance. Our work explicitly labels them for clarity.

Multiple GAN works use cGAN discriminator architecture (Miyato & Koyama, 2018). This architecture resembles CLIP (Radford et al., 2021), where the inner product between the visual embedding and the class embedding is computed at the end and maximized along with the adversarial objective. This is an implicit form of classifier guidance, as this architecture clearly does not apply to unconditional generation tasks.

GigaGAN proposes a matching loss (Kang et al., 2023), which trains $D$ to additionally evaluate class alignment. When training $G$, this implicitly encourages $G$ to generate samples that maximize classification. Hence, it is also an implicit form of classifier guidance.

## G. Gradient Normalization

Gradient normalization disentangles the scaling fluctuations that often occur as training progresses, and variations caused by the use of different discriminator architectures and gradient penalty strength.

For example, on the computation graph, we can capture the separate backward gradient norm from the adversarial objective and the optimal transport objective as received on the generator output. Without gradient normalization, the gradient norm of the adversarial objective as received from the discriminator changes during training (Figure 14).

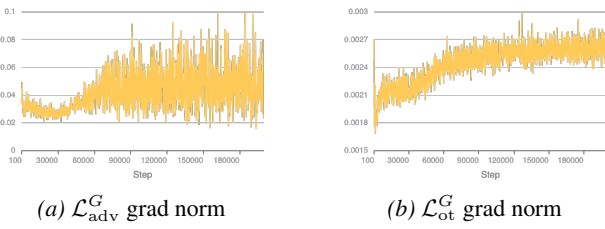

*(a) $\mathcal{L}_{\text{adv}}^{G}$ grad norm*      *(b) $\mathcal{L}_{\text{ot}}^{G}$ grad norm*

*Figure 14.* Without gradient normalization

With gradient normalization, the gradient norm of the adversarial objective is normalized to a constant scale and does not vary during training (Figure 15).

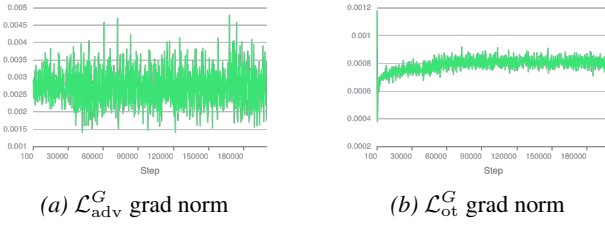

*(a) $\mathcal{L}_{\text{adv}}^{G}$ grad norm*      *(b) $\mathcal{L}_{\text{ot}}^{G}$ grad norm*

*Figure 15.* With gradient normalization

This disentanglement is beneficial for studying hyperparameters, but is not strictly necessary for achieving the best performance.

## H. Connections to Flow Models

Adversarial flow models are a type of discrete-time flow models. Samples can be transported between the data and prior distributions through the probability flow by solving the difference equation:

$$x_0 = x_1 + \sum_{i=1}^{S} \left( G(x_{\tau_i}, \tau_i, \tau_{i-1}) - x_{\tau_i} \right), \quad x_1 \sim \mathcal{Z}, \quad (39)$$

where the summation runs backward from $i = S$ to $i = 1$ with a total of $S$ sampling steps, and $\tau$ is a list of discrete timesteps satisfying $\tau_0 = 0$, $\tau_S = 1$.

# I. Implementation

The PyTorch (Paszke et al., 2019) implementation is provided in Algorithms 2 and 3 for reference.

---

**Algorithm 2** Adversarial Flow.

---

```
1  from torch.nn import functional as F
2
3
4  def interpolate(x, z, t):
5      t = t.view(-1, 1, 1, 1)
6      return (1 - t) * x + t * z
7
8
9  def training_step(
10     gen, dis,
11     gen_trainable_params, dis_trainable_params,
12     samples, conditions, noises_src, noises_tgt, noises_r1, noises_r2, timesteps_src, timesteps_tgt,
13     mode="dis", pred_type="x",
14     gp_scale=0.25, gp_bsz_ratio=0.25, gp_eps=0.01, cp_scale=0.01, ot_scale=0.2,
15     augment_fn, grad_norm_fn,
16     cg=None, noises_cg=None, timesteps_cg=None, cg_scale=0.003, cg_flow=True,
17 ):
18     assert mode in ["dis", "gen"]
19     assert pred_type in ["x", "v"]
20
21     for p in gen_trainable_params:
22         p.requires_grad_(mode == "gen")
23     for p in dis_trainable_params:
24         p.requires_grad_(mode == "dis")
25
26     samples_src = interpolate(samples, noises_src, timesteps_src)
27     samples_tgt = interpolate(samples, noises_tgt, timesteps_tgt)
28     samples_tgt_pred = gen(samples_src, conditions, timesteps_src, timesteps_tgt)
29
30     if pred_type == "v":
31         samples_tgt_pred = samples_src - (timesteps_src - timesteps_tgt).view(-1, 1, 1, 1) * samples_tgt_pred
32
33     samples_tgt_real_aug, samples_tgt_pred_aug = augment_fn(samples_tgt, samples_tgt_pred)
34     weighting = (timesteps_src - timesteps_tgt).abs().clamp_min(0.001)
35
36     if mode == "dis":
37         bsz = len(samples)
38         gp_bsz = max(round(bsz * gp_bsz_ratio), 1)
39         gp_scale = gp_scale * weighting[:gp_bsz] / gp_eps**2
40
41         samples_tgt_real_gp = samples_tgt_real_aug[:gp_bsz] + gp_eps * noises_r1[:gp_bsz]
42         samples_tgt_pred_gp = samples_tgt_pred_aug[:gp_bsz] + gp_eps * noises_r2[:gp_bsz]
43
44         logits_real = dis(samples_tgt_real_aug, conditions, timesteps_tgt)
45         logits_pred = dis(samples_tgt_pred_aug, conditions, timesteps_tgt)
46         logits_real_gp = dis(samples_tgt_real_gp, conditions[:gp_bsz], timesteps_tgt[:gp_bsz])
47         logits_pred_gp = dis(samples_tgt_pred_gp, conditions[:gp_bsz], timesteps_tgt[:gp_bsz])
48
49         dis_loss_adv = F.softplus(-(logits_real - logits_pred)).mean()
50         dis_loss_r1 = (logits_real_gp - logits_real[:gp_bsz]).square().mul(gp_scale).mean()
51         dis_loss_r2 = (logits_pred_gp - logits_pred[:gp_bsz]).square().mul(gp_scale).mean()
52         dis_loss_cp = (logits_real + logits_pred).square().mul(cp_scale).mean()
53
54         return dis_loss_adv + dis_loss_r1 + dis_loss_r2 + dis_loss_cp
55     else:
56         logits_pred = dis(grad_norm_fn(samples_tgt_pred_aug), conditions, timesteps_tgt)
57         logits_real = dis(samples_tgt_real_aug, conditions, timesteps_tgt)
58
59         gen_loss_adv = F.softplus(-(logits_pred - logits_real)).mean()
60         gen_loss_ot = (samples_tgt_pred - samples_src).square().mean([1,2,3]).mul(ot_scale / weighting).mean()
61         gen_loss_cg = 0.0
62
63         if cg is not None:
64             samples_tgt_pred_cg = (
65                 interpolate(samples_tgt_pred, noises_cg, timesteps_cg) if cg_flow else samples_tgt_pred
66             )
67             logits_cg = cg(samples_tgt_pred_cg, timesteps_cg)
68             gen_loss_cg = F.cross_entropy(logits_cg, conditions, reduction="none").mul(cg_scale).mean()
69
70         return gen_loss_adv + gen_loss_ot + gen_loss_cg
```

---

**Algorithm 3** Gradient Normalization.

```
1  import torch
2  import torch.distributed as dist
3
4
5  class GradientNormalization(torch.nn.Module):
6      def __init__(self, ema_decay=0.9, eps=1e-8, target_scale=1.0):
7          super().__init__()
8          self.ema_decay = ema_decay
9          self.eps = eps
10         self.target_scale = target_scale
11         self.register_buffer("square_avg", torch.tensor(0.0))
12
13     def forward(self, x):
14         return _GradientNormalizationFn.apply(
15             x,
16             self.square_avg,
17             self.ema_decay,
18             self.eps,
19             self.target_scale
20         )
21
22
23 class _GradientNormalizationFn(torch.autograd.Function):
24     @staticmethod
25     def forward(ctx, x, square_avg, ema_decay, eps, target_scale):
26         ctx.square_avg = square_avg
27         ctx.ema_decay = ema_decay
28         ctx.eps = eps
29         ctx.target_scale = target_scale
30         return x.clone()
31
32     @staticmethod
33     def backward(ctx, grad_output):
34         square_avg = ctx.square_avg
35         ema_decay = ctx.ema_decay
36         eps = ctx.eps
37         target_scale = ctx.target_scale
38
39         # Multiply n here is equivalent to divide by sqrt(n) later in the paper.
40         grad_sq_sum = grad_output.square().sum() * grad_output.numel()
41
42         # Here, we compute avg not sum for distributed training.
43         # This is only to exchange the local grad_sq_sum.
44         # We still want it to be in local scale, not global scale.
45         if dist.is_initialized():
46             dist.all_reduce(grad_sq_sum, op=dist.ReduceOp.AVG)
47
48         square_avg.lerp_(grad_sq_sum, 1 - ema_decay)
49         scale = square_avg.sqrt() + eps
50         grad_output = grad_output * (target_scale / scale)
51         return grad_output, None, None, None, None
```

