# OpenReview forum: "Adversarial Flow Models"
_ICML.cc/2026/Conference — ICML 2026 regular_

### Official Review · Reviewer_3hDq · 2026-03-10

**Soundness:** 3
**Presentation:** 3
**Significance:** 3
**Originality:** 3
**Overall Recommendation:** 4
**Confidence:** 5

**Summary:**

The paper proposes Adversarial Flow Models (AFM), a generative modeling framework that combines ideas from GANs and flow/flow-matching models.

**Compliance With Llm Reviewing Policy:**

Affirmed.

**Key Questions For Authors:**

1. Could the authors clarify what fundamentally distinguishes this approach from existing flow-matching or consistency-based generative models beyond the use of an adversarial objective?

2. The paper suggests that learning a deterministic optimal transport mapping can stabilize adversarial training. Could the authors provide additional intuition or theoretical insights explaining why this transport structure leads to improved stability compared to traditional GAN generators?

3. Does the deterministic flow parameterization reduce this sensitivity, or does the model still require similar tuning as standard GANs?

4. The results focus primarily on ImageNet-256. How well does the proposed framework scale to higher-resolution generation (e.g., 512 or 1024) or other modalities such as video, audio, or 3D generation?

5. Since the method aims to support one-step or few-step generation, how does the quality–efficiency trade-off compare with recent diffusion acceleration methods or consistency models?

6. Does the deterministic transport learned by the generator remain stable when the training distribution changes slightly (e.g., domain shift or noisy data)? Understanding this may provide insight into the robustness of the learned flow.

7. Since the model learns an explicit deterministic mapping from noise to data, is it possible to analyze or visualize the structure of this mapping to better understand how the model transports mass from the latent distribution to the data distribution?

8. Do the authors observe any differences in generalization behavior compared with standard GANs or diffusion models, particularly in terms of mode coverage or sample diversity?

9. Could the authors comment on whether the adversarial flow formulation could be combined with score-based or diffusion objectives during training to further improve stability or generation quality?

**Limitations:**

The paper proposes an interesting hybrid framework combining adversarial training with deterministic flow-based generation. The empirical results are strong and demonstrate competitive performance on large-scale benchmarks. However, the conceptual novelty relative to existing flow matching and adversarial training approaches may appear somewhat incremental, and the paper provides limited theoretical analysis of the proposed method.

**Strengths And Weaknesses:**

It provides strong empirical results and analyses.

---

> ### Author Rebuttal · Authors · 2026-03-27
>
> **Comparisons to consistency methods**
>
> Consistency models learn the deterministic probability flow ODE transport and enforce pointwise consistency of the flow map between source and target timesteps. The adversarial objective enforces distributional matching to the target timestep via a discriminator network. Adversarial approaches offer several benefits: (1) the generator can save capacity, as it does not need to model all timesteps; and (2) distributional matching empirically yields sharper results than pointwise matching. Originally, adversarial training was proposed as a standalone family of generative models, namely GANs. Our work establishes a connection between adversarial and flow-based model families, generalizes adversarial training to multi-step generation, and improves the training stability of the generator.
>
> **On optimal transport**
>
> Traditional adversarial objectives enforce only distributional matching, without constraining the underlying transport plan. As a result, the learned mapping can drift during training, making optimization unstable and less efficient. In contrast, our formulation encourages the generator to learn a specific transport map—namely, the Wasserstein-2 optimal transport from the noise distribution to the data distribution. We achieve this by introducing an additional optimal transport regularization term that minimizes the expected squared L2 transport cost: $\mathbb{E}_z || G(z)-z ||^2_2$. This regularization guides the model toward more structured and stable mappings. We refer to our response to Reviewer 9W1J for further discussion on the theoretical connections to classical optimal transport theory.
>
> **On deterministic flow reducing sensitivity**
>
> Yes, learning a deterministic flow is easier for optimization. Most of the prior GAN work relied on very specialized generator architectures to prevent drifting. When switching to standard transformer architecture, the training diverges (Table 1). By learning a deterministic transport, we are able to perform adversarial training on standard transformer for both the generator and the discriminator, thus reducing the sensitivity on architecture selection. However, the gradient penalty and the optimal transport loss scale are still hyperparameters that need to be tuned.
>
> **On ImageNet 256px evaluation**
>
> We refer to our response to Reviewer t1FA for further discussion on the evaluation criteria. Most prior and concurrent work on few-step generation reports results exclusively on the ImageNet dataset at 256px resolution, which has become a standard and widely accepted convention for methodology-focused research. Following this precedent, we leave the exploration of higher resolutions and other modalities to future work.
>
> **On quality and efficiency trade off and comparison with SOTA**
>
> Tables 4 and 5 demonstrate that our method achieves better FID scores compared to prior and concurrent methods in both standard single-step and multi-step settings. Our approach exhibits a similar quality-efficiency trade-off as consistency models, with multi-step generation still providing slightly better performance than single-step. Additionally, we investigate end-to-end training of 56-layer and 112-layer 1NFE models. These deeper models have the same FLOPs and wall-clock inference time as the 28-layer 2NFE and 4NFE models but achieve improved FID scores, highlighting a promising direction for more efficient high-quality generation.
>
> **On robustness to data distributions**
>
> Our proposed method is agnostic to the data distribution being generated, similar to flow matching and GANs. We have verified its robustness on ImageNet generation, and we expect it to generalize effectively to other modalities and data distributions.
>
> **On visualization of deterministic mapping**
>
> The deterministic mapping is visualized in Figure 1. In the 1D mixture of Gaussians case, our model learns a deterministic optimal transport, whereas GANs learn arbitrary mappings. For ImageNet, we provide visualization in Appendix B. In Figures 9a and 9b, we show that our model learns similar layout for the same seed compared to flow matching.
>
> **On generalization behavior**
>
> As elaborated in section 3.5, adversarial training can lead to different generalization behavior compared to flow matching due to the inclusion of a learned discriminator. We find our model yields better perceptual results compared to flow matching in the guidance-free setting (Figs 9a, 9b) and is supported by the FID (Tab 6). Mode coverage and sample diversity are not sacrificed as reflected in the FID. We also show latent interpolation in Figure 10.
>
> **On connections with score distillation**
>
> Similar to adversarial training, score distillation also only encourages the matching of the distribution without constraining the transport. Our optimal transport regularization can also be applied to encourage the learning a deterministic transport. We intend to explore this in future work.

---

> > ### Author Rebuttal · Reviewer_3hDq · 2026-04-05
> >
> > The authors have addressed some of my comments.

---

### Official Review · Reviewer_t1FA · 2026-03-12

**Soundness:** 3
**Presentation:** 4
**Significance:** 3
**Originality:** 2
**Overall Recommendation:** 4
**Confidence:** 4

**Summary:**

The paper introduces adversarial flow models, which train a generator with a standard relativistic adversarial loss combined with an L2 optimal transport regularizer (Eq. 12) to learn a deterministic noise-to-data mapping. The OT loss encourages the generator to recover the same transport plan as flow matching with linear interpolation, while the adversarial loss enforces distribution matching. A gradient normalization technique (Eq. 23) balances the two loss components across model scales. The framework supports both single-step and multi-step generation on standard DiT architectures. On class-conditional ImageNet-256 in a 32x32x4 VAE latent space, the XL/2 model reaches FID 2.38 (1NFE, with guidance) and 3.98 (1NFE, without guidance). Extra-deep 112-layer models trained end-to-end via depth repetition achieve FID 1.94 in a single step.

**Compliance With Llm Reviewing Policy:**

Affirmed.

**Final Justification:**

The authors have adequately addressed most of my concerns in their rebuttal. I maintain my score.

**Key Questions For Authors:**

1. Have you attempted any experiments at higher resolution (512+) or on text-to-image? What are the main barriers to scaling?
2. The 1NFE results in Tables 4 and 6 appear to use single-step trained models (no timestep projection, 673M params vs 675M for multi-step). Can you confirm this explicitly? If not, what is the performance if you train a single-step model?
3. What is the total wall-clock training cost (GPU hours) for AF-XL/2 to reach best FID, compared to AlphaFlow-XL/2 and MeanFlow-XL/2 under the same hardware?
4. How sensitive is the cosine decay endpoint for $\lambda\_\mathrm{ot}$ (Table 2 uses 0.2 to 0.01)? Was this schedule tuned per model size or shared across B/2 through XL/2?
5. What happens if you ablate the gradient normalization (Eq. 23) and instead manually tune $\lambda\_\mathrm{ot}$ for each model size? Is the normalization essential or just convenient?

**Limitations:**

The authors discuss memory overhead from $D$, the use of CG instead of CFG, and known limitations of adversarial discriminators (Section 4.4, Appendix E.1).

**Strengths And Weaknesses:**

Strengths:

- The no-guidance results are good. Table 6 shows AF-XL/2 at FID 3.98 (1NFE, no guidance), beating SiT-XL/2 at FID 8.30 with 250NFE. The explanation in Section 3.5 is helpful for understanding and Figure 9 backs this up visually. This is a genuinely useful property.

- The framework is simple and sits on top of unmodified DiT. This makes the results easy to reproduce and compare fairly against flow-matching baselines.

- The authors did a thorough ablation study. Tables 1-3 and Table 8 isolate the effects of $\lambda\_\mathrm{ot}$, $\lambda\_\mathrm{gp}$, OT decay, and guidance type. Table 1 is particularly informative: training diverges without OT loss regardless of $\lambda\_\mathrm{gp}$, confirming the regularizer is necessary and not just helpful.

Weaknesses:

- The core method amounts to adding an L2 penalty $\frac{1}{n}\\|G(z) - z\\|\_2^2$ to a fairly standard GAN setup. Adversarial distillation methods already combine adversarial losses with regression objectives; swapping the teacher prediction for the noise input $z$ and training from scratch is a natural simplification rather than a conceptual leap. The resulting method works well, but the novelty is more empirical than technical.

- Every experiment is on class-conditional ImageNet-256. This benchmark is important but saturated. What the field needs now is evidence that GAN-based methods can scale to higher resolutions, text-to-image generation, or other complex domains. The paper offers no such evidence. Even a preliminary experiment on ImageNet-512 or a small text-to-image setup would materially strengthen the contribution.

- Section 3.2 claims (line 172-173) that an infinitesimal $\lambda\_\mathrm{ot}$ breaks the symmetry and creates a unique global minimum. This is stated without proof and is unlikely to hold in general for non-convex min-max objectives. Figure 2 contradicts the spirit of the claim: in practice, $\lambda\_\mathrm{ot}$ too small fails to escape local minima, while $\lambda\_\mathrm{ot}$ too large collapses to identity. The decay schedule in Table 2 further shows the method is quite sensitive to how $\lambda\_\mathrm{ot}$ is annealed. The authors should either formalize the argument or reframe it as heuristic motivation.

- Training compute is 1.88x per update compared to MeanFlow (Appendix E.2), due to heavy regularization on $D$. The paper reports epoch counts but not wall-clock time or total GPU hours, making direct cost comparisons impossible. For a method targeting practical single-step generation, training efficiency matters.

- The paper bundles many engineering choices. Each is sensible on its own, but their interactions are not fully ablated. For instance, Table 8 ablates DA and CG, but there is no ablation isolating the effect of gradient normalization (Eq. 23) versus a simple fixed $\lambda\_\mathrm{ot}$ with manual tuning.

---

> ### Author Rebuttal · Authors · 2026-03-26
>
> **On ImageNet resolution and T2I generation**
>
> ImageNet generation at 256px resolution has become the standard benchmark for evaluating few-step generation on the DiT backbone. At the time of writing, both prior state-of-the-art methods [shortcut, meanflow, iMM] and concurrent works [alphaflow, TiM] report results exclusively at 256px resolution. Subsequent works published after our submission [drifting, improvedmeanflow, ESC] also continue to evaluate only at 256px.
>
> While a few earlier works [sCM, AYF] report results at 512px resolution, they are based on the EDM2 convolutional architecture rather than the DiT backbone. Therefore, we believe that reporting results at 256px resolution on the DiT backbone—consistent with the most relevant and recent prior work—is both appropriate and aligned with established conventions.
>
> We have conducted preliminary experiments on text-to-image (T2I) generation and observed promising results. In particular, we find that our Adversarial Flow Models can be stably trained for T2I tasks on standard transformer architectures without divergence. However, training a full-scale T2I model and achieving performance comparable to SOTA requires computational resources beyond the scope of this work. As our primary contribution is methodological, we follow established practice [shortcut, meanflow, iMM, alphaflow, sCM, drifting, ESC] and report results on ImageNet. We believe it is appropriate and aligned with established conventions to leave large-scale T2I training and further exploration to future work.
>
> Reference:
> * [DiT] Scalable Diffusion Models with Transformers
> * [shortcut] One Step Diffusion via Shortcut Models
> * [meanflow] Mean Flows for One-step Generative Modeling
> * [iMM] Inductive Moment Matching
> * [alphaflow] AlphaFlow: Understanding and Improving MeanFlow Models
> * [TiM] Transition Models: Rethinking the Generative Learning Objective
> * [improvedmeanflow] Improved Mean Flows: On the Challenges of Fastforward Generative Models
> * [drifting] Generative Modeling via Drifting
> * [ESC] On the Design of One-step Diffusion via Shortcutting Flow Paths
> * [sCM] Simplifying, Stabilizing and Scaling Continuous-Time Consistency Models
> * [AYF] Align Your Flow: Scaling Continuous-Time Flow Map Distillation
> * [EDM2] Analyzing and Improving the Training Dynamics of Diffusion Models
>
> **On training single-step models**
>
> Yes, we confirm that those models are trained using the single step objective only. We will clarify this in the final paper.
>
> **On the training wall-clock time**
>
> Our final XL/2 1NFE model was trained on 64xA100 40G GPUs in 3 days. The training can also be fitted on less GPUs (8x 80G). We measure the MeanFlow training speed using an open-source PyTorch implementation in our setup, and the total training time needed is approximated to be 2~2.5 days. This is consistent with our FLOPs analysis. Our model uses more compute but yields better FID than MeanFlow.
>
> **On the cosine decay endpoint**
>
> The ablation experiments are conducted on B/2 model only and the same setting is applied to models of other sizes. We provide additional ablation studies on different terminal OT under constant 1e-4 learning rate. Overall, the terminal OT scale does influence the final FID. The result reflects our expectation, where OT=0 leads to generator drifting while OT=0.1 is too large and compete with distributional matching.
>
> | OT | FID |
> |---|---|
> | 0.2 | 29.4 |
> | 0.2->0.1 | 10.56 |
> | 0.2->0.01 | 8.51 |
> | 0.2->0 | 8.69 |
>
> **On gradient normalization**
>
> Reviewer 9W1J has asked a similar question. Please refer to the newly attached experiment results there. We show that the normalization technique is only meant to provide a disentangled space for better hyperparameter search, but it is not strictly necessary. We show that it is possible to manually tune $\lambda_{ot}$ without gradient normalization and achieve comparable results.
>
> **Conclusion**
>
> We thank the reviewer for the detailed and constructive feedback. We hope our rebuttal has address all raised questions and concerns. We hope that, upon reflection, the reviewer finds our clarifications satisfactory and considers our work in a positive light for the final evaluation.

---

> > ### Author Rebuttal · Reviewer_t1FA · 2026-04-04
> >
> > The authors have adequately addressed most of my concerns in their rebuttal. I maintain my score.

---

### Official Review · Reviewer_9W1J · 2026-03-13

**Soundness:** 2
**Presentation:** 3
**Significance:** 3
**Originality:** 2
**Overall Recommendation:** 4
**Confidence:** 4

**Summary:**

This paper proposes Adversarial Flow Models (AF), a generative modeling framework intended to unify adversarial training (GAN-style) with flow/transport-based generation. The central idea is to keep the adversarial objective for high-fidelity distribution matching, while adding an optimal-transport-inspired regularizer that encourages the generator to learn a deterministic noise-to-data mapping (rather than an arbitrary transport plan), which the authors argue stabilizes adversarial training and aligns the learned transport with the linear-interpolation / squared-cost behavior commonly associated with flow-matching models. AF supports both native one-step generation (without needing intermediate-timestep training/consistency propagation) and multi-step generation via a time-conditioned formulation that can jump between timesteps on an interpolation-defined probability flow. The paper also introduces a gradient normalization mechanism to make the balance between adversarial gradients and the OT regularizer less sensitive to model scale, and discusses how to incorporate classifier guidance in a way that better mimics “accumulated” guidance effects along a flow via a time-conditioned classifier applied on interpolated samples.

**Compliance With Llm Reviewing Policy:**

Affirmed.

**Key Questions For Authors:**

- Under what assumptions does minimizing the adversarial objective subject to your OT cost recover a unique Monge map in your setting? Can you provide either (i) a clearer theoretical statement (with assumptions), or (ii) an empirical diagnostic showing that different initializations converge to essentially the same coupling/map (beyond seed-level qualitative determinism)?

- How much of the reported performance/stability depends on the proposed backward-pass gradient normalization operator? Please include an ablation (at least for one model size) showing training stability and final FID with and without it.

**Limitations:**

Yes.

**Strengths And Weaknesses:**

Strength
---
- Clear objective formulation and motivation. The paper identifies an important instability source in GANs—underspecification of the transport plan—and proposes to restrict the solution set by adding an OT-style penalty on the generator mapping, while retaining adversarial distribution matching.

- Empirical evidence that the OT term matters for stability. An ablation shows training diverges without the OT loss in their setting, and that too-small/too-large OT weight harms outcomes, motivating the scheduled decay.

- Novel synthesis of two paradigms. While adversarial objectives and transport/flow concepts are individually well-studied, the paper’s specific framing—using an OT-style cost to select a deterministic transport map while using adversarial training for distribution matching—feels like a meaningful conceptual combination.

- Practical contributions beyond the objective. Gradient normalization for stabilizing the adversarial-vs-OT gradient ratio, and guidance integration via time-conditioned classification on interpolated samples, are practical ideas likely to be reused/extended.

Weakness
---
- Theoretical claims are more heuristic than rigorous: 1) The argument that adding the OT term yields a “unique minimizer” is presented informally (e.g., relying on the OT loss’s unique minimizer at identity), but the combined constrained problem (match data distribution + minimize transport cost) is subtler; uniqueness depends on assumptions (existence/uniqueness of Monge map, absolute continuity, etc.). 2) The paper repeatedly asserts alignment with flow-matching optimal transport behavior; this is plausible in spirit but not proved, and it is unclear whether the training procedure truly recovers the same coupling/transport map as typical flow-matching training pipelines.

- Reliance on multiple “training tricks” beyond the core idea. Besides OT regularization, the strongest results appear to depend on a collection of stabilizers (gradient normalization operator, EMA reuse/reload, discriminator reload, discriminator augmentation in guided runs, etc.). The paper does not fully isolate which components are essential versus merely helpful, e.g., a targeted ablation that demonstrates necessity/benefit of the proposed gradient normalization mechanism.

- Scope of evaluation is narrow. The empirical story is almost entirely ImageNet 256 latent generation; it is unclear how well the approach generalizes to text-to-image, higher resolutions.

- Training cost and complexity may limit adoption. Even if inference is 1-step, training includes a full discriminator plus multiple regularizers/penalties and stabilizing procedures; the paper itself notes compute/memory drawbacks and that D-related issues remain a limitation.

---

> ### Author Rebuttal · Authors · 2026-03-26
>
> **On deterministic mapping**
>
> We thank the reviewer for pointing out the additional complexity in jointly enforcing distribution matching and transport cost minimization.
>
> The learning of a deterministic transport mapping in our framework is motivated by classical optimal transport theory under the following conditions:
> 1. The source distribution is absolutely continuous with respect to the Lebesgue measure (satisfied by using a Gaussian source in the single-step setting, and preserved in the multi-step setting because the interpolated samples are convolution of Gaussian).
> 2.  Both source and target distributions have finite second moments (satisfied by the data distributions to be generated in our experiments, and the Gaussian noise distribution).
> 3.  The transport cost is quadratic.
>
> Under these conditions, Brenier's theorem guarantees the existence of a unique optimal transport map between the source and target distributions.
>
> In our formulation, we restrict the transport plan to deterministic mappings parameterized by the generator network and minimize the expected quadratic displacement $\mathbb{E}_z ||G(z) - z||^2_2$. The generator is assumed to be sufficiently expressive to approximate this map.
>
> The GAN objective encourages matching between the generated and target distributions, and achieves exact marginal matching at the global optimum under standard assumptions (infinite capacity and optimal optimization). Within this setting, the quadratic regularization biases the solution toward the unique optimal transport map.
>
> Empirically, we observe that our method consistently learns the same deterministic mapping across different random initializations in the 1D mixture of Gaussians experiment (Fig. 1), whereas standard GANs produce different transport mappings. This provides evidence that the proposed objective promotes convergence toward a canonical transport solution. The code to reproduce this experiment will be published.
>
> We will remove the statement comparing to the transport behavior of flow matching, as establishing equivalence would require additional theoretical justification and is not central to our contribution.
>
> **On gradient normalization**
>
> As the paper has pointed out, the original intention of proposing gradient normalization is to have disentangled scaling control of the adversarial objective and the optimal transport objective. This is only to disentangle the potential scaling fluctuations that often happen as training progresses, and variations caused by the use of different discriminator architectures and gradient penalty strength, etc.
>
> For example, on the computation graph, we can capture the separate backward gradient norm from the adversarial objective and the optimal transport objective as received on the generator output. Without the gradient normalization, the gradient norm of the adversarial objective as received from the discriminator changes during training (shown in the graph below by the blue curve; this is the B/2 model training from scratch):
>
> * Figure 1: https://afmicml.blob.core.windows.net/file/grad_norm_off.png
>
> With gradient normalization, the gradient norm of the adversarial objective is normalized to a constant scale and does not vary during training (shown in the graph below by the red curve; same B/2 model training):
>
> * Figure 2: https://afmicml.blob.core.windows.net/file/grad_norm_on.png
>
> The disentanglement is beneficial to the study of the hyperparameters, but it is not strictly necessary. We ran additional B/2 1NFE training trials without using the gradient normalization technique and searched the hyperparameter in this setting. After the same 20 epochs as in the ablation study of Table 1, we find that under GP=0.25, OT=0.5, the model can also obtain an FID of 54.6, comparable to the best results with gradient normalization, with an FID of 53.9. Also, when not using gradient normalization, setting OT=0 also leads to divergence with an FID of 173.5, proving that the OT regularization is truly essential.
>
> **On the scope of evaluation**
>
> We have addressed a similar concern raised by reviewer t1FA. Evaluating DiT backbones at 256px resolution is an established convention in few-step generation research. Moreover, for work whose primary contribution is methodological, it is both common and appropriate to focus on ImageNet, while leaving large-scale T2I experiments to future work.
>
> **Conclusion**
>
> We hope our response addresses the concerns. We will edit the paper according to the feedback and attach the additional results presented in this rebuttal. We would also like to point out because our method enables single-step training only, our work also demonstrates the end-to-end training of specialized 56 and 112-layer 1NFE models. This sheds light on the effect of network depth on few-step generation. We hope these clarifications help demonstrate the significance and novelty of our work, and we look forward to the reviewers’ continued assessment.

---

> > ### Author Rebuttal · Reviewer_9W1J · 2026-04-03
> >
> > I appreciate the authors for addressing my concerns.
> > However, I remain concerned about the scope of the evaluation. While it is standard practice to evaluate few-step generation models on ImageNet-256, this alone is not sufficient to demonstrate the scalability of the proposed method. In that regard, have the authors considered demonstrating scalability through scaling-law analysis?

---

> > > ### Author Response · Authors · 2026-04-03
> > >
> > > We thank the reviewer for the response. We have demonstrated the scalability of our method by following the convention established by MeanFlow. We train our models on DiT-S/2 (130M), DiT-M/2 (306M), DiT-L/2 (457M), and DiT-XL/2 (673M) architectures. In Tables 4 and 6, the experiments show that the FID improves as the model size increases, and our method outcompetes MeanFlow across all model sizes. Table 9 of the appendix lists a detailed comparison of our method against prior and concurrent work across different scaling variations.
> > >
> > > In comparison, many prior and concurrent works [shortcut, imm, alphaflow] only report the B/2 and XL/2 settings. Our work specifically reports S/2, M/2, L/2, XL/2 to demonstrate the scaling property. We understand that the reviewer may be seeking even larger models. However, XL/2 is the largest model reported by the prior few-step generation work, and we follow the convention for fair comparisons. Furthermore, due to the stabilizing effect of learning deterministic transport, our method enables adversarial training on standard transformer architectures (DiT). We believe this is more favorable to scaling compared to prior adversarial work that relies on custom architectures.
> > >
> > > We hope our response addresses the reviewer's remaining concerns.

---

### Decision · Program_Chairs · 2026-04-30

**Decision:**

Accept (regular)

**Comment:**

This paper presents a new generative model that combines an adversarial loss with an L2 optimal transport regularizer to learn a deterministic noise-to-data mapping. The optimal transport loss yields the same transport plan as flow matching, whereas the adversarial loss enforces distributional matching. Gradient normalization balances the two loss components in a scale-invariant manner.

All reviews provide a positive evaluation of the paper.
- Rev. 9W1J appreciates the motivation, the generalization, and the complementarity of the two objectives, as well as the gradient normalization contribution. However, they find their claims more heuristic than theoretical, with multiple "tricks" to make it work. In addition, they find the evaluation on ImageNet-256 limited and an increased computational cost.
- Rev. appreciated the unaltered DiT model that allows for easy comparisons, the simplicity of the approach, and the thorough ablations. They are concerned about the novelty, as adversarial loss was already used in distillation approaches, the claims about a unique minimizer without a proof, the increased computational cost, and the evaluation limited to ImageNet-256.
- Rev. t1FA appreciated the strong empirical results and the detailed analysis. They are concerned about the evaluation being limited to ImageNet-256, incremental contribution, and limited theoretical analysis.

All three reviewers provided a very similar evaluation, with concerns about the limited theoretical analysis, the limited novelty, and the limited validation on ImageNet-256. The authors provided a rebuttal that answered most of the questions asked. All reviewers maintained their positive outlook for the paper, even though some concerns remained (e.g. the ImageNet-256 as the only evaluation).

Overall, I agree with the reviewers about the contribution being simple, the strong analysis, and the improved results. I agree that an additional dataset, image resolution, or task would have made the paper stronger. However, even in its current shape, I think the paper has some significant merits, and I will recommend it for acceptance.